# Large-Scale Multiway Clustering with Seeded Clustering

Jiaxin Hu [*]
Department of Statistics
University of Wisconsin-Madison

Multiway clustering methods for higher-order tensor observations have been developed in various fields, including recommendation systems, neuroimaging, and social networks. However, high computational costs hinder the applications of tensor-based approaches to real-world large-scale data. Here, we propose a large-scale multiway clustering framework under tensor block model, named LS-TBM, with accuracy guarantees. LS-TBM leverages seeded clustering to break down the expensive high-dimensional tensor clustering into two fast low-dimensional steps. Our two-step algorithm substantially reduces the time and space complexities from polynomial to logarithmic rates while maintaining the exact recovery of community structures, under certain signal conditions. We also establish the theoretical phase transition of LS-TBM performance with a key interplay between signal levels and seed sizes. Numerical experiments with synthetic data and real large-scale Uber Pickup data highlight LS-TBM's superior performance in practice.

## 1. Introduction

Multiway data has been extensively collected across various disciplines [1, 2]. Tensors, or multi-dimensional arrays, are considered as an effective tool to represent and analyze such multiway data with multiple indices. One important task in tensor analysis is multiway clustering. Applications of multiway clustering arise in a wide range of studies, including social relation graph clustering [3], brain network community detection in neuroscience [4], and hypergraph analysis [5]. Many tensor-based approaches [6–11] have been successfully developed to capture the higher-order structures and solve the multiway clustering task with theoretical guarantees. Particularly, *Tensor Block Model* (TBM, [7]) is one of the most popular statistical models for multiway clustering. TBM is considered as a higher-order generalization of matrix stochastic block model (SBM). The goal of TBM is to identify the underlying community structure in all directions (Output in Figure 1) from noisy tensor observations (Input in Figure 1). Previous works [7, 9] have investigated the theoretical properties and developed clustering algorithms with accuracy guarantees under TBM.

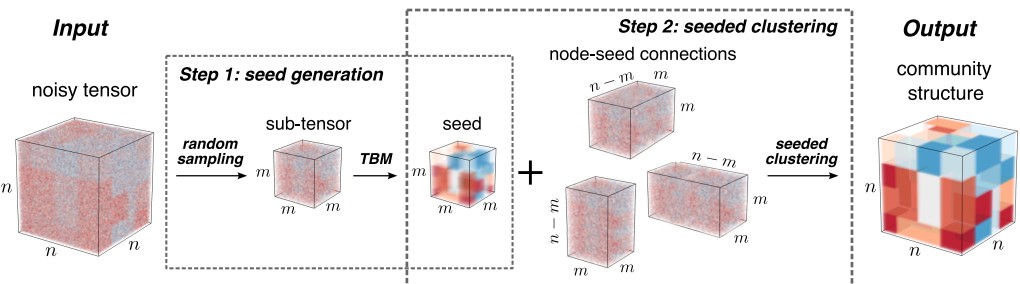

Figure 1: LS-TBM illustration for an order-3 tensor of with dimension $n$ and seed size $m$ on all modes.

Despite the rapid development of tensor methods, one significant drawback of most tensor-based algorithms, including the TBM algorithm in [9], is their computational inefficiency. Computational costs for one algorithm cover several aspects: (1) the storage size of inputs, (2) the *peak* memory

---

[*]Corresponding author: jhu267@wisc.edu

Second Conference on Parsimony and Learning (CPAL 2025).

cost to implement all intermediate calculations, and (3) the running time to obtain outputs. Due to multiple orders, the computational costs for tensor-based algorithms will inflate polynomially faster than one-dimensional and matrix-based algorithms. For example, the typical one-dimensional spectral clustering [12] for $n$ entities has time complexity $\mathcal{O}(n^2)$ while the higher-order spectral clustering [9] for $n$ entities on each of $K$ directions has time complexity $\mathcal{O}(n^{K+1})$. This computational obstacle brings financial and time concerns to apply tensor-based algorithms to large-scale data in practice, even though tensor methods possess better theoretical properties.

In this work, we provide a large-scale multiway clustering framework, named *Large-Scale TBM* (*LS-TBM*), that substantially reduces the computational burden while maintaining similar accuracy as full TBM clustering. The key idea of LS-TBM is the utilization of *seeded clustering*: given a subset of node community assignments, called *seed*, we are able to infer the community assignments of remaining nodes. Specifically, we divide the multiway clustering into two steps: seed generation and seeded clustering. In Step 1, we randomly sample the node sets and apply the full TBM algorithm on the sub-tensor associated with subsampled nodes to obtain the seed. In Step 2, we infer the community assignments of all remaining nodes by comparing the node-seed connections with the block means estimated by the seed and sub-tensor. See Figure 1 for illustration.

**Our contributions.** We summarize the main contributions with our proposed LS-TBM below.

1. Our LS-TBM substantially reduces the algorithm time and space complexities while maintaining the theoretical guarantees to exactly recover full community structures. Particularly, LS-TBM drops polynomial complexities to logarithmic complexities with a strong signal-to-noise ratio (SNR) level. Table 1 shows the leading performance of LS-TBM in computation.

2. We establish accuracy guarantees for the two steps and the overall LS-TBM. We present the phase transition of LS-TBM performance with a key theoretical interplay between the SNR level and seed size $m$ to the clustering accuracy. Figure 2 visualizes the phase transition and interplay.

3. We confirm the superior empirical performance of LS-TBM in large-scale numerical experiments.

| Algorithm | Full TBM [9] | LS-TBM | LS-TBM (Strong SNR) | LS-TBM (Weak SNR) |
|---|---|---|---|---|
| Time Complexity | $\mathcal{O}(n^{K+1})$ | $\mathcal{O}(m^{K+1})$ | $\mathcal{O}(\log^{(K+1)/(K-1)} n)$ | $\mathcal{O}(n^{(K+1)/2})$ |
| Space Complexity | $\mathcal{O}(n^K)$ | $\mathcal{O}(m^{K-1}n)$ | $\mathcal{O}(n \log n)$ | $\mathcal{O}(n^{(K-1)/2+1})$ |

Table 1: Time and space complexities of full TBM [9] and LS-TBM on an order-$K$ tensor with dimension $n$ and seed size $m$. See Sections 2, 3, and Remark 2 for detailed discussions.

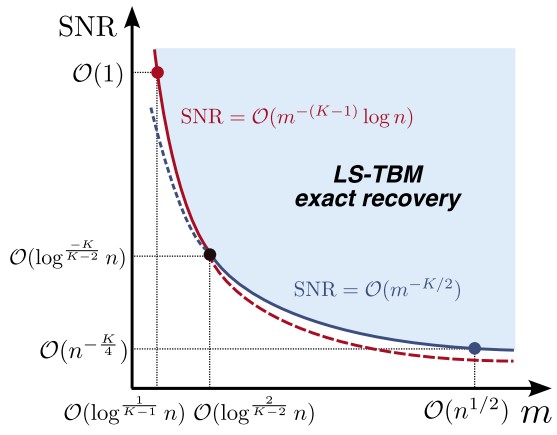

Figure 2: Phase transition of order-$K$ LS-TBM performance with a theoretical interplay between the SNR level and seed size $m$. As SNR reduces from strong $\mathcal{O}(1)$ to weak $\mathcal{O}(n^{-K/4})$, LS-TBM requires an increasing seed size $m$ from $\mathcal{O}(\log^{1/(K-1)} n)$ to $\mathcal{O}(n^{1/2})$ to the achieve exact recovery. When the combination ($m$, SNR) lies in the light blue area, LS-TBM fully recovers the community structure with a high probability.

**Related work.** Here, we review several relevant lines of literature for comparison. The first line is about multiway clustering algorithms. One prominent approach is higher-order spectral clustering [8, 9, 11, 13–15], which leverages low-dimensional spectral representations of data tensors. Higher-order spectral clustering extends traditional spectral clustering methods and incorporates various tensor decomposition methods such as CAN-DECOMP/PARAFAC (CP) decomposition [16] and Tucker decomposition [17]. Specific strategies have also been developed for certain scenarios, including tensor-SCORE [8], angle-based iterations [10] for degree-corrected TBM, and the generalized linear model with alternating iterations [15] for binary tensor observations. Only a few works consider the computational difficulties in the applications with large-scale tensors while our work pays special attention to the computational feasibility, resource

requirement, and running time in implementation. Through our framework, real applications with massive tensor data are able to practically enjoy the methodological achievements obtained by previous multiway clustering literature.

The second line focuses on scalable community detection. Several large-scale methods have been proposed for one-dimensional or matrix community detection problems, including fast pseudo-likelihood approaches [18, 19], label propagation algorithms [20, 21], divide-and-conquer strategies [22–25], and subsampling techniques [26, 27]. However, these methods rarely address the multiway clustering problem for higher-order tensors. Given the unique algebraic properties of tensors, such extensions are non-trivial, and our work addresses this gap with a scalable tensor method.

The last line of research relates to the seed expansion. Generally, seeds refer to objects that require special attention in a system, and seed expansion propagates information from the seed to the entire system. In community detection, seed expansion reveals the full structure via node-seed connections through random walks [28, 29] or by comparing node characterizations [30, 31]. This concept has also been applied to other areas, including graph matching [32, 33], essential node identification [34], and relation extraction [35]. To the best of our knowledge, LS-TBM is the first work to adapt seed expansion to multiway clustering via seeded clustering.

**Notation.** We use lower-case letters (e.g., $a$) for scalars, lower-case bold letters (e.g., $\boldsymbol{a}$) for vectors, upper-case letters (e.g., $S$) for sets of integers, upper-case bold letters (e.g., $\boldsymbol{M}$) for matrices, and calligraphy letters (e.g., $\mathcal{X}$) for higher-order tensors with order equal or larger than 3. One exception is that we reserve letters 'c' (e.g., $c, c_1$) and 'C' (e.g., $C, C_1$) for generic small or large positive constants, respectively. For a positive integer $n$, let $[n] = \{1, \ldots, n\}$. For a set $S$, let $|S|$ denote the cardinality and $S(i)$ denote the $i$-th element in $S$. For two sequences $\{a_n, b_n\}$ of positive numbers, we denote $a_n \lesssim b_n$ or $a_n = \mathcal{O}(b_n)$ if $\lim_{n \to \infty} a_n / b_n \leq c$ and $a_n \gtrsim b_n$ or $a_n = \Omega(b_n)$ if $\lim_{n \to \infty} a_n / b_n \geq c$ for some constant $c > 0$; we denote $a_n = o(b_n)$ if $\lim_{n \to \infty} a_n / b_n = 0$; and we denote $a_n \asymp b_n$ if $a_n \gtrsim b_n$ and $a_n \lesssim b_n$. We use $\mathbb{1}\{\cdot\}$ for the 0-1 indicator function, $\|\cdot\|_2$ for the $\ell_2$ norm, $\langle \cdot, \cdot \rangle$ for the inner product, $\otimes$ for the Kronecker product, $\boldsymbol{1}_n \in \mathbb{R}^n$ for the vector with all elements 1, $\boldsymbol{I}_n \in \mathbb{R}^{n \times n}$ for the identity matrix, and $\text{diag}(\boldsymbol{v}) \in \mathbb{R}^{n \times n}$ for the diagonal matrix with elements in a vector $\boldsymbol{v} \in \mathbb{R}^n$. Consider an order-$K$ $(n_1, \ldots, n_K)$-dimensional tensor $\mathcal{X} \in \mathbb{R}^{n_1 \times \cdots \times n_K}$ with entries $\mathcal{X}(i_1, \ldots, i_K)$ for $i_k \in [n_k], k \in [K]$. We use colon ' : ' as a shorthand representing all entries in a particular direction. For example, $\mathcal{X}(i_1, \ldots, i_{K-1}, :) \in \mathbb{R}^{n_K}$ is a tensor fiber, and $\mathcal{X}(i_1, :, \ldots, :) \in \mathbb{R}^{n_2 \times \cdots \times n_K}$ is a tensor slice of the first mode. For a matrix $\boldsymbol{X}$, $\boldsymbol{X}_{i,:}$ and $\boldsymbol{X}_{:,j}$ are the $i$-th row and $j$-th column, respectively. We use $\text{Mat}_k(\cdot)$ for tensor unfolding operation along the $k$-th mode and $\text{Mat}_l(\mathcal{X}) \in \mathbb{R}^{n_l \times \prod_{k \neq l} n_k}$. Consider node subsets $S_k \in [n_k]$ for $k \in [K]$. We define sub-tensor associated with $\{S_k\}_{k \in [K]}$ as

$$\mathcal{X}(S_1, \ldots, S_k) = [\![\mathcal{X}(S_1(j_1), \ldots, S_K(j_K))]\!]_{j_k \in [|S_k|], k \in [K]} \in \mathbb{R}^{|S_1| \times \cdots \times |S_K|}.$$

The multilinear multiplication of a core tensor $\mathcal{C} \in \mathbb{R}^{r_1 \times \cdots \times r_K}$ by matrices $\boldsymbol{M}_k \in \mathbb{R}^{n_k \times r_k}$ results in an order-$K$ $(n_1, \ldots, n_K)$-dimensional tensor $\mathcal{X}$, denoted as $\mathcal{X} = \mathcal{C} \times_1 \boldsymbol{M}_1 \times_2 \cdots \times_K \boldsymbol{M}_K$, with entries

$$\mathcal{X}(i_1, \ldots, i_K) = \sum_{j_1, \ldots, j_K} \mathcal{C}(j_1, \ldots, j_K) \boldsymbol{M}_1(i_1, j_1) \cdots \boldsymbol{M}_K(i_K, j_K).$$

Last, we drop the subscript $k \in [K]$ for any sequence with $K$ elements $\{n_k\}_{k \in [K]}$ and use following shorthands for $\{n_k\}$: $n_* = \prod_{k \in [K]} n_k, n_{-l} = \prod_{k \in [K], k \neq l} n_k, \bar{n} = \max_{k \in [K]} n_k, \underline{n} = \min_{k \in [K]} n_k$.

## 2. Large-Scale Tensor Block Model Framework

Before the LS-TBM algorithm, we recall the TBM [7] formula for setup. Consider an order-$K$ tensor observation $\mathcal{Y} \in \mathbb{R}^{n_1 \times \cdots \times n_K}$. Suppose that there are $r_k$ communities on the $k$-th mode of $\mathcal{Y}$. Let $z_k \colon [n_k] \mapsto [r_k], k \in [K]$ denote the community assignment functions. We say $\mathcal{Y}$ follows the TBM if

$$\mathcal{Y} = \mathcal{X} + \mathcal{E}, \quad \text{where} \quad \mathcal{X} = \mathcal{C} \times_1 \boldsymbol{M}_1 \times_2 \cdots \times_K \boldsymbol{M}_K, \tag{1}$$

$\mathcal{X} = \mathbb{E}[\mathcal{Y}] \in \mathbb{R}^{n_1 \times \cdots \times n_K}$ is the mean signal tensor, $\mathcal{E} \in \mathbb{R}^{n_1 \times \cdots \times n_K}$ is a noise tensor whose entries are independent and zero-mean with variance upper bounded by $\sigma^2$, $\mathcal{C} \in \mathbb{R}^{r_1 \times \cdots \times r_K}$ is the core tensor

collecting the block means among communities, and $\boldsymbol{M}_k \in \{0,1\}^{n_k \times r_k}, k \in [K]$ are membership matrices such that $\boldsymbol{M}_k(i, a) = \mathbb{1}\{z_k(i) = a\}$. The community assignments $\{z_k\}$ or $\{\boldsymbol{M}_k\}$ are main parameters of interests. We focus on high dimensional cases where $n_k$'s increase while $r_k = \mathcal{O}(1)$.

The key idea of LS-TBM to alleviate computational burden is dividing the *high-dimensional and expensive* full algorithm into two *low-dimensional and cheap* steps: seed generation and seeded clustering. For a better narrative, we firstly introduce seeded clustering and then discuss seed generation.

## 2.1. Seeded clustering

Seeded clustering aims to extend the seed, i.e., the partial community assignments, to the full community structure. Taking the seed as "true labels", seeded clustering in fact transfers the unsupervised clustering problem to a relatively easy, supervised classification task.

Specifically, for illustration, we consider the seeded clustering on the first mode with a perfect seed $\{\tilde{z}_k : S_k \mapsto [r_k], S_k \subset [n_k]\}$ such that $\tilde{z}_k(i) = z_k(i)$ and $|S_k| = m_k$ for all $i \in S_k, k \in [K]$. Based on formula (1), the TBM structure still holds on the sub-tensor associated with $\{S_k\}$

$$\mathcal{X}(S_1, \ldots, S_K) = \mathcal{C} \times_1 \boldsymbol{M}_1(S_1, :) \times_2 \cdots \times_K \boldsymbol{M}_K(S_K, :),$$

where $\boldsymbol{M}_k(S_k, :) \in \{0,1\}^{m_k \times r_k}$ are membership matrices corresponding to $\{\tilde{z}_k\}$. By tensor multiplication, we have

$$\boldsymbol{C}_1 = \boldsymbol{W}_1^T \boldsymbol{X}_1(S_1, S_2 \times \cdots \times S_K)[\boldsymbol{W}_2 \otimes \cdots \otimes \boldsymbol{W}_K]^T, \tag{2}$$

where $\boldsymbol{C}_1 = \mathrm{Mat}_1(\mathcal{C}) \in \mathbb{R}^{r_1 \times r_{-1}}, \boldsymbol{X}_1(S_1, S_2 \times \cdots \times S_K) = \mathrm{Mat}_1(\mathcal{X}(S_1, \ldots, S_K)) \in \mathbb{R}^{m_1 \times m_{-1}}$ are mode-1 matricizations of the core tensor and the sub-tensor, respectively, and

$$\boldsymbol{W}_k = \boldsymbol{M}_k(S_k, :)(\mathrm{diag}(\boldsymbol{1}_{m_k}^T \boldsymbol{M}_k(S_k, :)))^{-1}, \quad k \in [K],$$

are weighted membership matrices such that $\boldsymbol{W}_k^T \boldsymbol{M}_k(S_k, :) = \boldsymbol{I}_{r_k}$. The matrix $\boldsymbol{C}_1$ has $r_1$ unique rows, and the row $\boldsymbol{C}_1(a, :)$ collects the connections between the $a$-th community on the first mode to all $r_{-1}$ communities on the other modes. Notice that, for an arbitrary $j \in [n_1]/S_1$, we have

$$\boldsymbol{C}_1(z_1(j), :) = \boldsymbol{x}_j := \boldsymbol{X}_1(j, S_2 \times \cdots \times S_K)[\boldsymbol{W}_2 \otimes \cdots \otimes \boldsymbol{W}_K]^T,$$

where $\boldsymbol{X}_1(j, S_2 \times \cdots \times S_K) = \mathrm{Mat}_1(\mathcal{X}(j, S_2, \ldots, S_K)) \in \mathbb{R}^{r_{-1}}$ collects the *node-seed connections* between the node $j$ and seed node subsets $S_2, \ldots, S_K$. Then, we are able to obtain the community assignment for node $j$, $z_1(j)$, by comparing the aggregated node-seed connection vector $\boldsymbol{x}_j$ with $r_1$ rows of $\boldsymbol{C}_1$. We recognize our seeded clustering as a supervised procedure. Instead of calculating the pairwise similarities among feature vectors $\boldsymbol{x}_j$'s, we classify the nodes via the similarities between $\boldsymbol{x}_j$'s and $r_1$ "true" reference centroids in $\boldsymbol{C}_1$. This supervised nature makes seeded clustering more computationally efficient than typical unsupervised clustering (e.g., $k$-means) who calculates similarities between all pairs of feature vectors. In practice, we use the noisy observation and imperfect seeds for estimation. See detailed procedures in the Sub-algorithm 2 of Algorithm 1.

## 2.2. Seed generation

Seed generation aims to obtain the partial community assignments $\{\tilde{z}_k : S_k \mapsto [r_k], S_k \subset [n_k]\}$. Seed provides a low-dimensional sketch of the whole community structure and will serve as the "true labels" for the following seeded clustering. As a pivot linking two steps, seed plays a critical role that determines the final clustering accuracy. A poor-quality seed misaligned with true $z_k$ can mislead the assignments for remaining nodes. The misalignment in the seed will even be exaggerated in the full assignments due to the large dimension, making it more difficult to achieve overall exact recovery. A high-quality seed should satisfy three conditions: 1. $\{S_k\}$ are representative enough to cover all $\{r_k\}$ communities, 2. $\{\tilde{z}_k\}$ are accurate enough, and 3. sizes of the seed $\{m_k\}$ are proper.

Condition 1 is straightforward. It is impossible to obtain an accurate estimate of $\boldsymbol{C}_1(a, :)$ in (2) if none of $a$-th community members is in $S_1$. Fortunately, the *uniform sampling* addresses this requirement under most cases. If the true community structure is balanced, $\{S_k\}$ from uniform sampling are able to cover all communities with a high probability. See Section 3.2 for the theoretical proof.

---

**Algorithm 1** Large-scale multiway clustering under tensor block model (LS-TBM)

---
**Sub-algorithm 1: Seed generation**

---
**Input:** Observation $\mathcal{Y}$, number of communities $\{r_k\}$, sizes of the seed $\{m_k\}$, number of iterations $T$, relaxation factor in $k$-means: $M > 1$ ($T, M$ are hyperparameters only for Algorithm 2).

  1: Uniformly sample the node subsets $S_k$ from $[n_k]$ such that $|S_k| = m_k$, for all modes $k \in [K]$.
  2: Apply full TBM Algorithm 2 with $\{r_k\}, T, M$ on the sub-tensor $\mathcal{Y}(S_1, \ldots, S_K) \in \mathbb{R}^{m_1 \times \cdots \times m_K}$.

**Output:** Seed $\{\tilde{z}_k : S_k \mapsto [r_k]\}$ output by Algorithm 2.

---
**Sub-algorithm 2: Seeded clustering**

---
**Input:** Observation $\mathcal{Y}$, seed $\{\tilde{z}_k : S_k \mapsto [r_k]\}$ from Sub-algorithm 1.

  3: Obtain the estimated core tensor with the seed $\{\tilde{z}_k\}$ as

$$\tilde{\mathcal{C}} = \mathcal{Y}(S_1, \ldots, S_K) \times_1 \tilde{\boldsymbol{W}}_1^T \times_2 \cdots \times_K \tilde{\boldsymbol{W}}_K^T, \text{ where } \tilde{\boldsymbol{W}}_k = \tilde{\boldsymbol{M}}_k(S_k, :)(\text{diag}(\mathbf{1}_{m_k}^T \tilde{\boldsymbol{M}}_k(S_k, :)))^{-1},$$

     and $\tilde{\boldsymbol{M}}_k(S_k, :) \in \{0, 1\}^{m_k \times r_k}$ are membership matrices spanned by $\tilde{z}_k$ for $k \in [K]$.
  4: **for** $k = 1$ to $K$ **do**
  5:    Let $S_k^c = [n_k]/S_k$ be the complement of $S_k$. Calculate the aggregated tensor observation

$$\mathcal{A}_k = \mathcal{Y}(S_1, \ldots, S_{k-1}, S_k^c, S_{k+1}, \ldots, S_K) \times_1 \tilde{\boldsymbol{W}}_1^T \times_2 \cdots \times_{k-1} \tilde{\boldsymbol{W}}_{k-1}^T \times_{k+1} \tilde{\boldsymbol{W}}_{k+1}^T \times_{k+2} \cdots \times_K \tilde{\boldsymbol{W}}_K^T.$$

  6:    Calculate matricizations $\tilde{\boldsymbol{C}}_k = \text{Mat}_k(\tilde{\mathcal{C}}) \in \mathbb{R}^{r_k \times r_{-k}}$ and $\boldsymbol{A}_k = \text{Mat}_k(\mathcal{A}_k) \in \mathbb{R}^{(n_k - m_k) \times r_{-k}}$. The rows of $\tilde{\boldsymbol{C}}_k$ serve as $r_k$ reference community centroids, and rows of $\boldsymbol{A}_k$ serve as aggregated feature vectors of remaining $n_k - m_k$ nodes waiting for classification.
  7:    **for** $j \in S_k^c$ **do**
  8:        Obtain the assignment for $j$ as $\hat{z}_k(j) = \arg\min_{a \in [r_k]} \|\tilde{\boldsymbol{C}}_k(a, :) - \boldsymbol{A}_k(j, :)\|_2^2$.
  9:    **end for**
10:    Combining seed assignments, obtain the full assignment $\hat{z}_k$ such that $\hat{z}_k(i) = \tilde{z}_k(i)$ for $i \in S_k$.
11: **end for**

**Output:** Estimated full community assignments $\{\hat{z}_k : [n_k] \mapsto [r_k]\}$.

---

Another advantage of uniform sampling is that no extra prior calculation is needed. Other sampling schemes [36, 37] are sophisticated but lead extra computation burdens. Condition 2 also ensures the estimation accuracy of $\boldsymbol{C}_1$ in (2). To achieve a high accuracy for seed assignments, we leverage the optimal full TBM algorithm [9] on the low-dimensional sub-tensor. We recall the algorithm of [9] as Algorithm 2 in Appendix. Condition 3 is the trickiest one and seed sizes $\{m_k\}$ are the key hyperparameters of the LS-TBM framework. If we choose large $m_k$'s, the computational burden will increase due to the usage of full TBM on the sub-tensor. Whereas, small $m_k$'s will lead the failure of Conditions 1 and 2 simultaneously. The choice of $\{m_k\}$ relates to the computational consideration, the signal level in $\mathcal{C}$, and the LS-TBM accuracy. For the question "how to choose proper $\{m_k\}$", we defer the discussion to Sections 3 and 4. Full procedures of LS-TBM framework are in Algorithm 1.

**Remark 1** (Complexity of LS-TBM Algorithm 1)**.** We analyze both time and space complexities, emphasizing the algorithm operation speed and memory cost, respectively. Consider the balanced case where $n_k \asymp n, m_k \asymp m$ for all $k \in [K]$ and assume $\{r_k\}, K$ as constants. The time complexity of Algorithm 1 is $\mathcal{O}(m^{K+1} + m^K + m^{K-1}n)$. The term $\mathcal{O}(m^{K+1})$ comes from the application of Algorithm 2 in Line 2. The other terms $\mathcal{O}(m^K + m^{K-1}n)$ come from the estimation of $\mathcal{C}$ (Line 3) and the aggregation (Line 5) in the seeded clustering step. The space complexity of Algorithm 1 is $\mathcal{O}(m^K + m^{K-1}n)$, dominated by the storage of the sub-tensor $\mathcal{Y}(S_1, \ldots, S_k)$ (Line 3) and node-seed connections $\mathcal{Y}(S_1, \ldots, S_k^c, \ldots, S_K)$ (Line 5). See comparison with full TBM in Table 1.

## 3. Theoretical Guarantees

We start the theoretical analysis with several definitions and a general assumption. First, we define

the clustering evaluation metrics. Take mode-1 as an example. The *misclassification error* of $\hat{z}_1$ is

$$\ell(\hat{z}_1, z_1) = \frac{1}{n_1} \min_{\pi \in \Pi} \sum_{i \in [n_1]} \mathbb{1}\{\hat{z}_1(i) \neq \pi \circ z_1(i)\},$$

where $\pi : [r_1] \mapsto [r_1]$ is a label permutation function, $\Pi$ collects all possible permutations on $[r_1]$, and $\circ$ denotes the composition operation. We say the estimate $\hat{z}_1$ exactly recovers the true assignment $z_1$ if $\ell(\hat{z}_1, z_1) < 1/n_1$. Another metric is the *misclassification loss* of estimate $\hat{z}_1$, which is defined as

$$L(\hat{z}_1, z_1) = \frac{1}{n_1} \min_{\pi \in \Pi} \sum_{i \in [n_1]} \mathbb{1}\{\hat{z}_1(i) \neq \pi \circ z_1(i)\} \|\text{Mat}_1(\mathcal{C})(\pi \circ z_1(i), :) - \text{Mat}_1(\mathcal{C})(\hat{z}_1(i), :)\|_2^2,$$

where $\mathcal{C}$ is the core tensor in TBM (1). We drop the true assignment in misclassification error and loss, e.g., $\ell(\hat{z}_1)$ and $L(\hat{z}_1)$, for simplicity. Second, we define the *signal-to-noise ratio* (*SNR*) of TBM as

$$\text{SNR} = \frac{\Delta_{\min}^2}{\sigma^2}, \quad \text{where } \Delta_{\min}^2 = \min_{k \in [K]} \Delta_k^2, \text{ and } \Delta_k^2 = \min_{a \neq b \in [r_k]} \|\text{Mat}_k(\mathcal{C})(a, :) - \text{Mat}_k(\mathcal{C})(b, :)\|_2^2,$$

and $\sigma^2$ is the variance upper bound for the independent entries in the noise tensor $\mathcal{E}$ of (1).

**Assumption 1** (Balanced communities)**.** *There exist universal positive constants $\alpha_1, \alpha_2$ such that $\alpha_1 n_k/r_k \leq \sum_{i \in [n_k]} \mathbb{1}\{z_k(i) = a\} \leq \alpha_2 n_k/r_k$, for all $a \in [r_k], k \in [K]$.*

Assumption 1 is mild. Such assumption is common in multiway clustering literature [7–10]. Practically, we ignore this balance assumption in the numerical experiments in Section 4.

## 3.1. Accuracy of seeded clustering

The accuracy of seeded clustering relies on the seed. Next assumption requires the seed balance.

**Assumption 2** (Balanced communities in seed node subsets)**.** *There exist universal positive constants $\alpha_3, \alpha_4$ such that $\alpha_3 m_k/r_k \leq \sum_{i \in S_k} \mathbb{1}\{z_k(i) = a\} \leq \alpha_4 m_k/r_k$, for all $a \in [r_k], k \in [K]$.*

Let $z_k^c : [n_k]/S_k \mapsto [r_k]$ denote the community assignments for remaining nodes, for all $k \in [K]$. Now, we are ready to present the accuracy of seeded clustering.

**Theorem 3.1** (Accuracy of seeded clustering)**.** *Suppose that the tensor observation $\mathcal{Y}$ follows TBM (1) with number of communities $\{r_k\}$. Let $\{\tilde{z}_k : S_k \mapsto [r_k]\}$ be the given seed, and let $\{\hat{z}_k : [n_k] \mapsto [r_k]\}$ denote the output of Sub-algorithm 2 in Algorithm 1 with $\mathcal{Y}$ and $\{\tilde{z}_k\}$. Suppose that Assumptions 1 and 2 hold. If SNR and the seed satisfy following conditions with some positive large constant $C$ and small constant $c$:*

$$SNR \geq C \frac{\bar{m}}{m_*} \log \bar{m}, \quad L(\tilde{z}_k) \leq c \frac{\Delta_{\min}^2}{\sqrt{\bar{m}}}, \quad \text{for } k \in [K], \tag{3}$$

*we have*

$$\ell(\hat{z}_k^c) \leq C' SNR^{-1} \exp\left(-c' m_{-k} SNR\right), \quad \text{for } k \in [K],$$

*with probability at least $1 - \exp(-c'' \underline{m}) - \exp(-c''' m_{-k} \Delta_{\min}^2/\sigma^2)$ and $C', c', c'', c'''$ are positive constants.*

Consider the balanced case where $n_k \asymp n$ and $m_k \asymp m$ for all $k \in [K]$. The first condition in (3) requires the SNR level to be at least $\mathcal{O}(m^{-(K-1)} \log m)$. This signal requirement aligns with the SNR requirement in the guarantee of the higher-order Lloyd algorithm [9] with a tensor of dimension $m$. Such condition indicates that our seeded clustering shares a similar spirit as the Lloyd iteration. The second condition in (3) provides the boundary of the seed's accuracy to guarantee a good performance in seeded clustering. Note that we do not assume the seed generation approach for Theorem 3.1. As long as the seed satisfies the condition (3) and other assumptions hold, the seeded clustering achieves a fast exponential error rate $\mathcal{O}(\exp(-m^{K-1}))$ given a fixed SNR.

## 3.2. Quality of seed generation

As described in Section 2.2 and indicated by Theorem 3.1, we consider the seed quality from two aspects: the coverage of community structure and the accuracy of seed assignments.

**Theorem 3.2** (Balanced communities in uniformly sampled node subsets). *Consider the node subsets $\{S_k\}$ obtained by uniform sampling from $\{[n_k]\}$, such that $|S_k| = m_k$ for all $k \in [K]$. Suppose that Assumption 1 holds. With probability at least $1 - C\exp(-cm_k)$, we have*

$$\frac{\alpha_3 m_k}{r_k} \leq \sum_{i \in S_k} \mathbb{1}\{z_k(i) = a\} \leq \frac{\alpha_4 m_k}{r_k}, \quad \textit{for all } a \in [r_k], k \in [K]$$

*where $C, c, \alpha_3, \alpha_4$ are some universal positive constants.*

By Theorem 3.2, we formally show that the simple uniform sampling selects node subsets with balanced true community structures. That is, node subsets $\{S_k\}$ generated by the Line 1 in Algorithm 1 satisfies Assumption 2, which is required by the seeded clustering guarantee Theorem 3.1.

Next, for accuracy, we present the theoretical guarantee in [9] as a corollary under our context.

**Corollary 1** (Accuracy of seed assignments [9]). *Suppose that the tensor observation $\mathcal{Y}$ follows TBM (1) with number of communities $\{r_k\}$. Let $\{\tilde{z}_k : S_k \mapsto [r_k]\}$ denote the output of Sub-algorithm 1 in Algorithm 1 with given inputs $\mathcal{Y}, \{m_k\}, T$, and $M$. Suppose that Assumption 1 holds. If SNR and number of iterations $T$ satisfy the following conditions with some positive large constants $C, C'$:*

$$SNR \geq Cm_*^{-1/2}, \quad T \geq C'\log \bar{m}, \tag{4}$$

*we have*

$$L(\tilde{z}_k) \leq C''\sigma^2 \exp\left(-\frac{c_1 m_*}{\bar{m}}SNR\right) + c_2\frac{\Delta_{\min}^2}{2^T} \leq c\Delta_{\min}^2/\sqrt{\bar{m}}, \quad \textit{for all } k \in [K], \tag{5}$$

*with a high probability at least $1 - \exp(-c'\underline{m}) - \exp(-c''m_{-k}\Delta_{\min}^2/\sigma^2)$. Here, $c, c', c'', c'', c_i$ are some small positive constants related to $M$ and $C''$ is a large positive constant.*

Under the balanced case, the first condition in (4) requires a stronger SNR level at rate $\mathcal{O}(m^{-K/2})$, compared with Theorem 3.1. Such stronger SNR requirement is due to the spectral initialization in full TBM Algorithm 2. The second condition in (4) requires a large number of iterations for the iterative Lloyd step in Algorithm 2. Note that the upper bound (5) aligns with condition (3) in Theorem 3.1. Combining with Theorem 3.2, we show that our seed generation, Sub-algorithm 1 in Algorithm 1, is good enough to provide high-quality seeds for seeded clustering.

## 3.3. Overall accuracy of large-scale multiway clustering

Now, we show the overall accuracy of Algorithm 1.

**Corollary 2** (Overall accuracy of LS-TBM). *Suppose that the tensor observation $\mathcal{Y}$ follows TBM (1) with number of communities $\{r_k\}$. Let $\{\hat{z}_k\}$ denote the output of Algorithm 1 with given inputs $\mathcal{Y}, \{m_k\}, T$, and $M$. Suppose that Assumption 1 holds. If SNR and number of iterations $T$ satisfy the following conditions with some positive large constants $C_1, C_2, C_3$:*

$$SNR \geq C_1 m_*^{-1/2} \vee C_2 \frac{\bar{m}}{m_*}\log \bar{m}, \quad T \geq C_3 \log \bar{m}, \tag{6}$$

*we have*

$$\ell(\hat{z}_k) \leq C_4 SNR^{-1} \exp\left(-\frac{c_1 m_*}{\bar{m}}SNR\right) + c_2 \frac{m_k}{2^T n_k}, \quad \textit{for all } k \in [K] \tag{7}$$

*with high probability at least $1 - C_5 \exp(-c_3\underline{m}) - \exp(-c_4\frac{m_*}{\bar{m}}SNR)$ as $m_k \to \infty, n_k \to \infty$ and some positive constants $c_i$'s and $C_i$'s.*

Corollary 2 combines the results in seeded clustering and seed generation. Conditions (6) for SNR and number of iterations are inherited from the seed generation step, Corollary 1. The error rate in (7) can be obtained with Theorem 3.1 and Corollary 1 by the fact that $\ell(\hat{z}_k) = \frac{m_k}{n_k}\ell(\tilde{z}_k) + \frac{n_k - m_k}{n_k}\ell(\hat{z}_k^c)$.

**Remark 2** (Interplay between seed size $m$ and SNR to the exact recovery of LS-TBM). Corollary 2 indicates the mutual effects of the seed size $m$ and SNR level to the misclassification error bound (7). Under the balanced case $n_k \asymp n, m_k \asymp m$, we investigate the interplay between $m$ and SNR to the exact recovery of LS-TBM to answer the question "*how to select a proper $m$*" raised in Section 3.2. Here, we assume $K \geq 3$ and $T$ is large enough to ignore the second term in the bound (7).

To achieve exact recovery with LS-TBM, i.e., $\ell(\hat{z}_k) \leq 1/n$ for all $k \in [K]$, we need following conditions to be fulfilled simultaneously by Corollary 2:

$$\text{SNR} \gtrsim m^{-K/2}, \quad \text{and} \quad m^{K-1}\text{SNR} \gtrsim \log n, \tag{8}$$

where the first inequality comes from (6) and the second inequality comes from (7). Based on (8), when $m \gtrsim \log^{2/(K-2)} n$, we need SNR $\gtrsim m^{-K/2}$ to achieve exact recovery; when $m \lesssim \log^{2/(K-2)} n$, exact recovery requires SNR $\gtrsim m^{-(K-1)} \log n$. Inequalities (8) also indicate the trade-off between $m$ and SNR: a larger SNR level is needed if we want to use a smaller $m$ in LS-TBM for multiway clustering, and the TBM with a smaller SNR requires a larger seed size $m$ for LS-TBM to recover. Figure 2 visualizes such phase transition of LS-TBM and the interplay (8). Besides the intersection point $\mathcal{O}(\log^{2/(K-2)} n)$, we also discuss two representative cases with strong and week SNRs.

First, consider the strong signal case with constant SNR $\asymp 1$. By the interplay (8), we only need a logarithmic seed size $m = \mathcal{O}(\log^{1/(K-1)} n)$ to achieve exact recovery. Then, based on Remark 1 and [9], the time complexity to fully recover the community structure with full TBM, $\mathcal{O}(n^{K+1})$, dramatically drops to $\mathcal{O}(\log^{(K+1)/(K-1)} n)$ with LS-TBM. Also, the space complexity drops from $\mathcal{O}(n^K)$ to $\mathcal{O}(n \log n)$. This complexity comparison proves that LS-TBM has a huge potential to release the computational burden of multiway clustering while keeping the exact recovery performance.

Second, we consider a relatively weak signal case with SNR $\asymp n^{-K/4}$. We need a polynomial seed size $m = \mathcal{O}(n^{1/2})$ for exact recovery by (8). Under this case, time complexity drops from $\mathcal{O}(n^{K+1})$ to $\mathcal{O}(n^{K+1}/2)$ with LS-TBM, and space complexity reduces from $\mathcal{O}(n^K)$ and $\mathcal{O}(n^{(K-1)/2+1})$. We conclude that LS-TBM polynomially relieves the computational burden even under a weak SNR.

Table 1 summarizes the complexity comparison under both strong and weak SNR cases.

# 4. Numerical Experiments

## 4.1. Simulations

For simulations, we consider order-3 tensor observations from TBM (1) with $n_k = n, r_k = r$ and seed size $m_k = m$ for $k = 1, 2, 3$. We generate true assignments $\{z_k\}$ via random sampling from $[r]$ and set noise level $\sigma^2 = 1$. We use the clustering error rate (CER), i.e., one minus the adjusted Rand index, to measure clustering accuracy. The CER is equal to the misclassification error $\ell(\hat{z})$ up to a constant factor [38]. We report average statistics and standard deviations across 30 replications [1].

Our first experiment verifies the theoretical interplay in Section 3. We choose three SNR levels from strong to weak: $\mathcal{O}(1), \mathcal{O}(\log^{-3} n)$, and $\mathcal{O}(n^{-3/4})$. Based on Remark 2, the seed size thresholds are $m = \mathcal{O}(\log^{1/2} n), \mathcal{O}(\log^2 n)$, and $\mathcal{O}(n^{1/2})$, respectively. LS-TBM performance is expected to change rapidly around the thresholds. Figure 3 confirms such phase transition. Under the strong SNR $= \mathcal{O}(1)$ case, all LS-TBM algorithms converge except that with constant seed size $m = \mathcal{O}(1)$; under the weak SNR $= \mathcal{O}(n^{3/4})$, a significant performance gap is observed between $m = \mathcal{O}(\log^2 n)$ and $\mathcal{O}(n^{1/2})$. Additional experiments in Appendix further support the rapid LS-TBM performance changes around the seed size threshold $m = \mathcal{O}(n^{1/2})$ under the weak SNR case.

Our second experiment compares the empirical performance of LS-TBM and full-TBM from accuracy (in CER), memory cost (peak RAM, in Mb), and running time (in second). We consider the strong SNR $= \mathcal{O}(1)$ cases. Experiments that exceed the memory limit 25 Gb will be accounted as infeasible. Figure 4 shows the leading advantage of LS-TBM in saving computational resources

---
[1]LS-TBM software is available at `https://github.com/Marchhu36/LS-TBM`.

without accuracy sacrifices. The memory cost and running time of full TBM inflate at a polynomial rate, while the costs of LS-TBMs increase in an extremely slow rate. Such phenomenon agrees with the complexity comparison in Table 1. Moreover, the gap in computational cost between LS-TBM and full TBM increases significantly as the order $K$ increases. See Appendix for the additional experiments with $K = 4, 5$. Therefore, we conclude that LS-TBM is favorable than full TBM for large-scale multiway clustering.

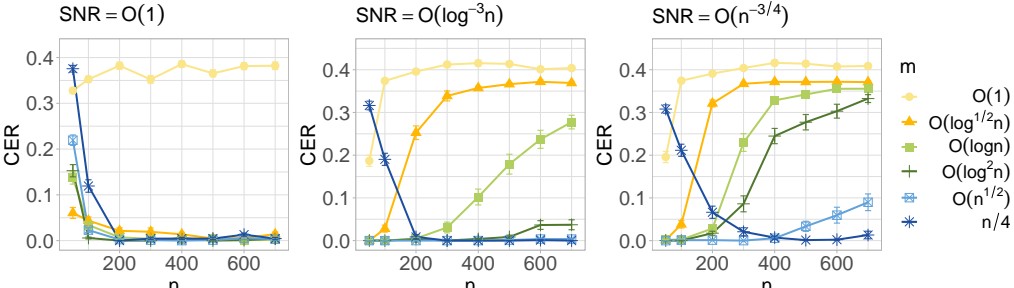

Figure 3: Interplay between the SNR and seed size $m$ in LS-TBM. Set dimension $n$ from 50 to 700, SNR $= \mathcal{O}(1), \mathcal{O}(\log^{-3} n), \mathcal{O}(n^{-3/4})$ and seed size $m = \mathcal{O}(1) \mathcal{O}(1), \mathcal{O}(\log^{1/2} n), \mathcal{O}(\log n), \mathcal{O}(\log^2 n), \mathcal{O}(n^{1/2}), n/4$.

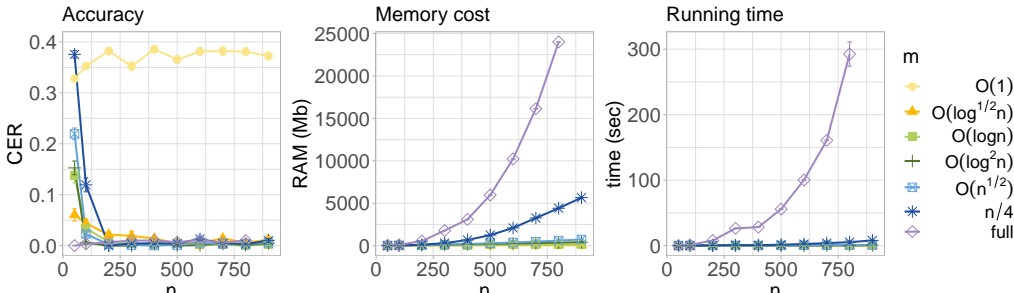

Figure 4: Empirical comparison between LS-TBMs and full TBM from accuracy (CER), memory cost (peak RAM, in Mb), and running time (in second). Set dimension $n$ from 50 to 900, SNR $= \mathcal{O}(1)$, and seed size $m$ from $\mathcal{O}(1)$ to $n/4$. Full TBM exceeds the memory upper limit of 25 Gb when $n > 800$.

## 4.2. Real data analysis

We apply our LS-TBM to the Uber Pickup Data [39] in New York City (NYC) from April 2014 to August 2014. We organize the data as an order-3 tensor $\mathcal{Y} \in \mathbb{R}^{4392 \times 445 \times 559}$ with count entries recording the hourly amount of Uber pickups, where mode-1 refers to timestamps of $183$ days $\times$ $24$ hours, mode-2 and mode-3 refer to 445 latitude and 559 longitude coordinates, respectively. To mimic the application on personal laptop, we choose $m_1 = 0.1n_1, (m_2, m_3) = 0.5(n_2, n_3)$ for LS-TBM, taking peak RAM around 5 Gb and running time 25 seconds. Due to the large dimension, it is time-consuming to perform $r$ selection on the entire $\mathcal{Y}$. We select $r_1 = 6$ based on the time combinations (Workday, Weekend) $\times$ (Early Morning, Daytime, Evening) and select $r_2 = r_3 = 4$ as the maximal numbers not leading singletons in LS-TBM clustering. For comparison, we still conduct the expensive full TBM with peak RAM around 27 Gb and running time 40 minutes.

Figure 5A shows the similar geographic community structure of NYC obtained by TBM and LS-TBM. The averaged CER between LS-TBM and full TBM on the last two modes (latitude and longitude) is 0.015. Particularly, both LS-TBM and full TBM identify an unique cluster highlighting the Manhattan district. For the first mode (timestamp), LS-TBM learns different time clusters compared to full TBM but captures major patterns. See detailed time cluster comparison in Appendix. Figure 5B shows three representative common time clusters: Cluster 1 (Early Morning $\times$ Weekdays), Cluster 2 (Daytime $\times$ Workdays), and Cluster 5 (Evening $\times$ Later Workdays). Meanwhile, by Figure 5C, the averaged Uber pickup amount in Manhattan neighborhoods increases from early morning to evening. Overall, given the similar clustering performance but the huge gap in com-

putational requirements, we say LS-TBM is more practically useful than full TBM for real-world large-scale multiway clustering tasks.

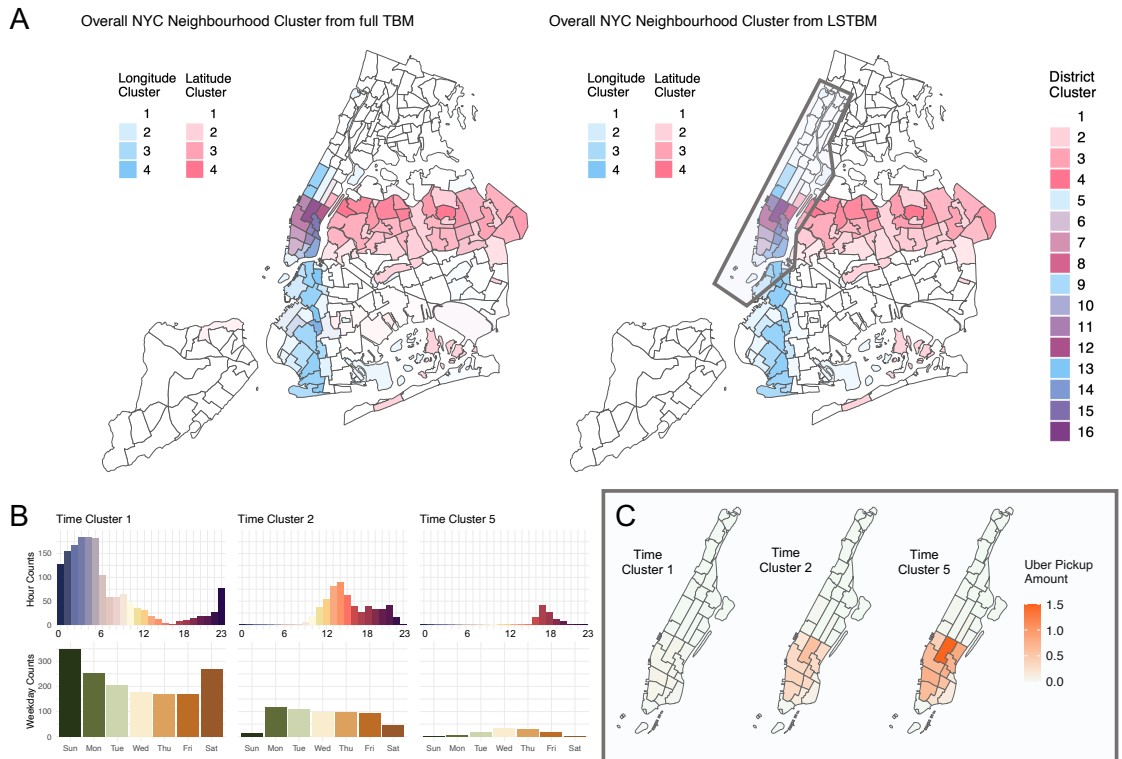

Figure 5: Multiway clustering on Uber Pickup Tensor. **A.** Geographic communities obtained by full TBM and LS-TBM. NYC is partitioned in 16 district clusters combined from 4 longitude clusters × 4 latitude clusters. **B.** Counts of 24 hours and 7 weekdays for timestamps in Time Clusters 1, 2, and 5 from LS-TBM. **C.** Estimated hourly Uber pickup amount (estimated $\mathcal{C}$ of (1)) in Manhattan neighborhoods under different time clusters.

# 5. Conclusion

This paper introduces an efficient multiway clustering framework, LS-TBM, for large-scale tensors. LS-TBM decomposes the high-dimensional and expensive TBM algorithm into two low-dimensional and cheap steps, seed generation and seeded clustering. Complexity analysis and accuracy guarantees theoretically proves the efficacy of LS-TBM: substantially reduces the computational cost while achieving a desirable clustering performance. In particular, under the strong SNR case, LS-TBM is able to exactly recover the community structure with logarithmic complexity, in contrast to the polynomial complexity required by the full TBM algorithm. Simulations and real data analysis using Uber Pickup data further empirically validate the superior efficiency of LS-TBM in terms of runtime and memory usage while maintaining a comparable accuracy to the full TBM algorithm.

LS-TBM is a generalizable framework with potential extensions in multiple directions. In the seed generation step, when additional prior community information is available, structural sampling may be preferable to uniform sampling for generating high-quality seeds. Furthermore, since the performance of LS-TBM only depends on seed quality, more sophisticated multiway clustering methods can be incorporated for seed generation in specific scenarios, such as cases with high sparsity and heavy-tailed observations. In the seeded clustering step, alternative distance metrics may be employed in Step 8 of Sub-algorithm 2 for different purposes. For instance, an angle-based comparison can be adopted for seeded clustering under the degree-corrected model, while people may use absolute distance for robust clustering. Therefore, we believe that LS-TBM serves as a strong foundation to the development of generalized multiway clustering methodologies for large-scale tensors.

## Acknowledgments

I thank Dr. Miaoyan Wang and Dr. Joshua Cape from the University of Wisconsin-Madison for their insightful feedback. I also thank Xinran Miao, Runshi Tang, Kwangmoon Park from the University of Wisconsin-Madison, as well as Haiming Ning, for their valuable discussions on real data analysis, paper presentation, and visualization. Additionally, I appreciate the anonymous reviewers for their helpful comments and suggestions.

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

# Appendix

# A. Full TBM Algorithms in [9]

We present the full TBM algorithm in [9] for consistency.

---

**Algorithm 2** Full TBM algorithm with spectral initialization and Lloyd iteration [9]

---

**Input:** Observation $\mathcal{Y}$, number of communities $\{r_k\}$, relaxation factor $M > 1$, iteration number $T$

**Higher-order spectral initialization**

1: Compute $\tilde{U}_k = \text{SVD}_{r_k}(\text{Mat}_k(\mathcal{Y}))$ for $k \in [K]$.
2: **for** $k = 1$ to $K$ **do**
3:     Compute the singular space estimator $\hat{U}_k$ via

$$\hat{U}_k = \text{SVD}_{\min\{r_k, r_{-k}\}} \left( \text{Mat}_k(\mathcal{Y} \times_1 \tilde{U}_1^T \times_2 \cdots \times_{k-1} \tilde{U}_{k-1}^T \times_{k+1} \tilde{U}_{k+1}^T \times_{k+2} \cdots \times_K \tilde{U}_k^T) \right)$$

4: **end for**
5: **for** $k = 1$ to $K$ **do**
6:     Calculate $\hat{Y}_k = \hat{U}_k \hat{U}_k^T \text{Mat}_k \left( \mathcal{Y} \times_1 \hat{U}_1^T \times_2 \cdots \times_{k-1} \hat{U}_{k-1}^T \times_{k+1} \hat{U}_{k+1}^T \times_{k+2} \cdots \times_K \hat{U}_k^T \right)$
7:     Find $z_k^{(0)} \colon [n_k] \mapsto [r_k]$ and centroids $\hat{x}_1, \ldots, \hat{x}_{r_k} \in \mathbb{R}^{r_{-k}}$ such that

$$\sum_{j \in [n_k]} \|\hat{Y}_k(j,:)^T - \hat{x}_{z_k^{(0)}(j)}\|_2^2 \leq M \min_{x_1, \ldots, x_{r_k} \in \mathbb{R}^{r_{-k}}, z_k \colon [n_k] \mapsto [r_k]} \sum_{j \in [n_k]} \|\hat{Y}_k(j,:)^T - x_{z_k(j)}\|_2^2.$$

8: **end for**
9: Obtain the spectral initialization $\{z_k^{(0)}\}$.

**Higher-order Lloyd algorithm**

10: **for** $t = 0$ to $T - 1$ **do**
11:     Update the block means $\mathcal{S}^{(t)}$ via

$$\mathcal{S}^{(t)}(i_1, \ldots, i_K) = \text{Average} \left( \left\{ \mathcal{Y}(j_1, \ldots, j_K) : z_k^{(t)}(j_k) = i_k, k \in [K] \right\} \right).$$

12:     **for** $k = 1$ to $K$ **do**
13:         **for** $j = 1$ to $n_k$ **do**
14:         Calculate $\mathcal{Y}_k^{(t)} \in \mathbb{R}^{r_1 \times \cdots \times r_{k-1} \times r_{k+1} \times \cdots \times r_K}$ such that

$$\mathcal{Y}_k^{(t)}(i_1, \ldots, i_{k-1}, j, i_{k+1}, \ldots, i_K) =$$
$$\text{Average} \left( \left\{ \mathcal{Y}(j_1, \ldots, j_{k-1}, j, j_{k+1}, \ldots, j_K) : z_l^{(t)}(j_l) = i_l, l \in [K]/k \right\} \right).$$

15:         Update the mode-$k$ membership for the $j$-th entry $z_k^{(t+1)}(j)$ via

$$z_k^{(t+1)}(j) = \underset{a \in [r_k]}{\arg\min} \|\text{Mat}_k(\mathcal{Y}_k^{(t)})(j,:) - \text{Mat}_k(\mathcal{S}^{(t)})(a,:)\|_2^2.$$

16:         **end for**
17:     **end for**
18: **end for**

**Output:** Estimated block memberships $\{z_k^{(T)}\}$.

---

# B. Additional Numerical Experiments

## B.1. Additional Simulation

Here, we provide additional simulation results for supplement.

The first additional experiments further investigate the LS-TBM under weak SNR $\mathcal{O}(n^{-3/4})$ cases. Based on Remark 2, $m = \mathcal{O}(n^{1/2})$ is the phase transition threshold for exact recovery, and LS-TBM performance is expected to change rapidly around this threshold. We consider two other choices for $m$, $\mathcal{O}(n^{1/2} \log^{-1/4} n)$ and $\mathcal{O}(n^{1/2} \log^{1/4} n)$, which are slightly smaller and larger than $\mathcal{O}(n^{1/2})$. Figure 6 shows that the LS-TBM accuracy indeed changes rapidly around the threshold $\mathcal{O}(n^{1/2})$, given that the actual seed sizes associated with the three $m$ values are very close. Moreover, Figure 2 demonstrates the divergent error for seeds as the dimension increases. This phenomenon implies that the divergent error of LS-TBM with $m$ equal to or smaller than $\mathcal{O}(n^{1/2})$ is caused by the TBM error on the sub-tensor. Such divergence agrees with the unstable performance of TBM at the phase transition thresholds, as reported in Han et al. [9, Figure 3]. TBM provides a stable performance only when SNR exceeds the threshold. Nevertheless, above numerical experiments still support the theoretical phase transition in Corollary 2, as LS-TBM exhibits sharp changes around the threshold $m = \mathcal{O}(n^{1/2})$.

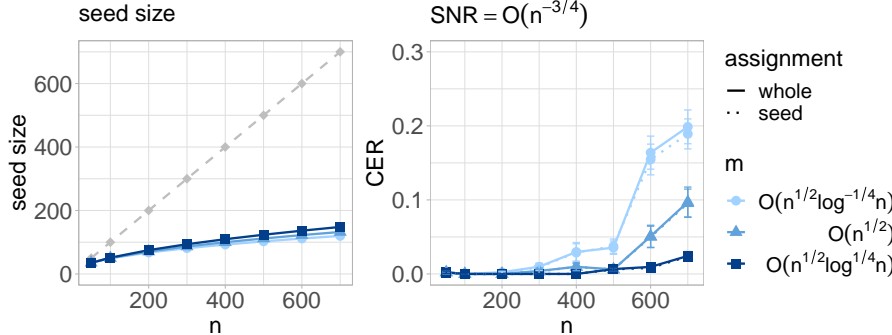

Figure 6: Additional theoretical verifications of LS-TBM under weak SNR $\mathcal{O}(n^{-3/4})$ cases. In addition to the phase transition threshold $m = \mathcal{O}(n^{1/2})$, we consider two other choices of $m$ for LS-TBM, $\mathcal{O}(n^{1/2} \log^{-1/4} n)$ and $\mathcal{O}(n^{1/2} \log^{1/4} n)$, which are slightly smaller and larger than $\mathcal{O}(n^{1/2})$. (Left) Actual seed sizes associated with different $m$'s. The gray dashed line refers to the dimension $n$. (Right) CER trajectories as the dimension $n$ from 50 to 700. CER trajectories for seeds (dotted lines) are largely overlapped with those for the LS-TBM Algorithm 1 outputs (solid lines).

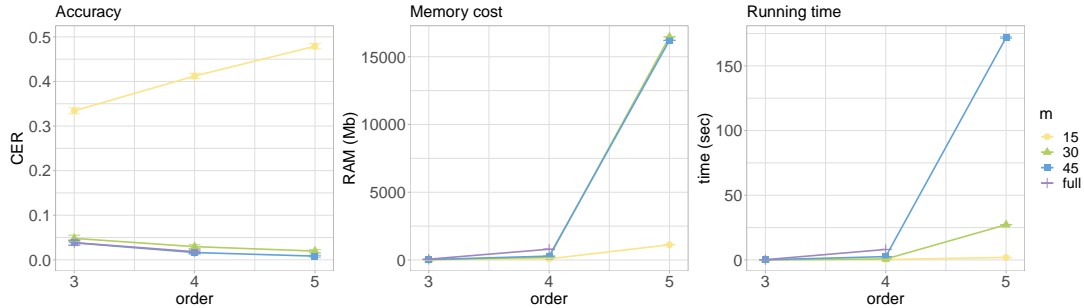

Figure 7: Empirical comparison between LS-TBMs and full TBM from accuracy (CER), memory cost (peak RAM, in Mb), and running time (in second). We consider dimension $n = 60$, SNR $= \mathcal{O}(1)$, tensor order $K = 3, 4, 5$, and seed size $m = 15, 30, 45$. Full TBM exceeds the memory upper limit of 32 Gb when $K = 5$.

The second additional experiment explores the empirical LS-TBM performance on tensors of order $K = 3, 4, 5$. Since the number of tensor entries grows exponentially with increasing $K$, we set $n = 60$

to ensure that the data tensor can be generated within 32 GB of memory. We also consider fixed seed sizes $m = 15, 30, 45$ for LS-TBM due to the small $n$. Figure 7 confirms that the computational costs for tensor methods increase exponentially as the tensor order increases. This result aligns with the complexity analysis in Table 1. Furthermore, this experiment indicates the importance of our scalable LS-TBM approach for the analysis of higher-order tensors.

## B.2. Additional results in Uber Pickup data analysis

Figure 8 summarizes the Time Cluster results of LS-TBM with $m_1 = 0.1n_1, (m_2, m_3) = 0.5(n_2, n_3)$ and full TBM for the Uber Pickup data. The presented LS-TBM results are obtained from a randomly selected run among multiple executions of the LS-TBM. The CER between the LS-TBM and TBM time assignments is 0.33.

Despite the relatively large CER, LS-TBM captures several key patterns consistent with TBM:

1. Time Cluster 1: The largest cluster with the smallest average Uber pickup amounts, primarily covering early morning hours from midnight to 6 a.m..

2. Time Clusters 2 and 3: Show more even timestamp distributions throughout the daytime.

3. Time Clusters 4, 5 and 6: Smaller clusters capturing evening hours, with Cluster 4 focusing on weekdays, while Clusters 5 and 6 concentrate around weekends. These clusters exhibit relatively higher pickup amounts than others.

To further evaluate LS-TBM, we consider the identification of Time Cluster 1, labeling other clusters as 0. Using TBM assignments as the ground truth, LS-TBM achieves an accuracy of 0.7, recall of 0.6, and specificity of 0.76. McNemar's test for the confusion matrix yields a p-value of 0.25, indicating no significant difference between LS-TBM and TBM assignments for Cluster 1. This pattern similarity is consistent across multiple runs of LS-TBM. The average Cluster 1 identification accuracy is 0.74 over 10 replications. Hence, we conclude that LS-TBM effectively captures the major clustering patterns of TBM for the Uber Pickup data.

Table 2 compares the computational performance between LS-TBMs with different $m$'s and TBM in Uber Pickup application. LS-TBM is shown to have a better computation performance than TBM. Additionally, Table 2 indicates the variability of LS-TBM across runs, with a CER variance of approximately 0.05. Note that TBM keeps the same result across different runs. The LS-TBM variability arises from the inherent randomness in the method, which is a trade-off for its computational speed-up. We leave the extension of LS-TBM incorporating more robust procedures as a future work.

| | $a = 0.1$ | $a = 0.2$ | $a = 0.3$ | $a = 0.5$ | $a = 0.7$ | Full TBM |
|---|---|---|---|---|---|---|
| Averaged CER | 0.18 (0.02) | 0.15 (0.02) | 0.12 (0.05) | 0.12 (0.02) | 0.08 (0.05) | - |
| mode-1 CER | 0.36 (0.03) | 0.34 (0.02) | 0.27 (0.09) | 0.32 (0.05) | 0.19 (0.15) | - |
| mode-2 CER | 0.11 (0.04) | 0.06 (0.03) | 0.06 (0.07) | 0.02 (0.01) | 0.03 (0.06) | - |
| mode-3 CER | 0.07 (0.03) | 0.05 (0.02) | 0.04 (0.04) | 0.02 (0.01) | 0.01 (0.01) | - |
| Peak RAM (Mb) | 721.26 (13.10) | 2858.20 (0.00) | 7643.19 (0.03) | 11716.60 (0.00) | 15869.99 (2116.02) | 26853.1 |
| Running time (sec) | 1.31 (0.10) | 9.05 (0.14) | 32.55 (0.18) | 188.61 (1.36) | 632.74 (1.98) | 2335.7 |

Table 2: Computational comparison between LS-TBM and TBM in real Uber Pickup tensor application. We set $a = m_k/n_k$ from 0.1 to 0.7 for LS-TBM and take full TBM assignments as reference "true" assignments in CER. Standard deviations across 10 replications for LS-TBM applications are recorded in parentheses.

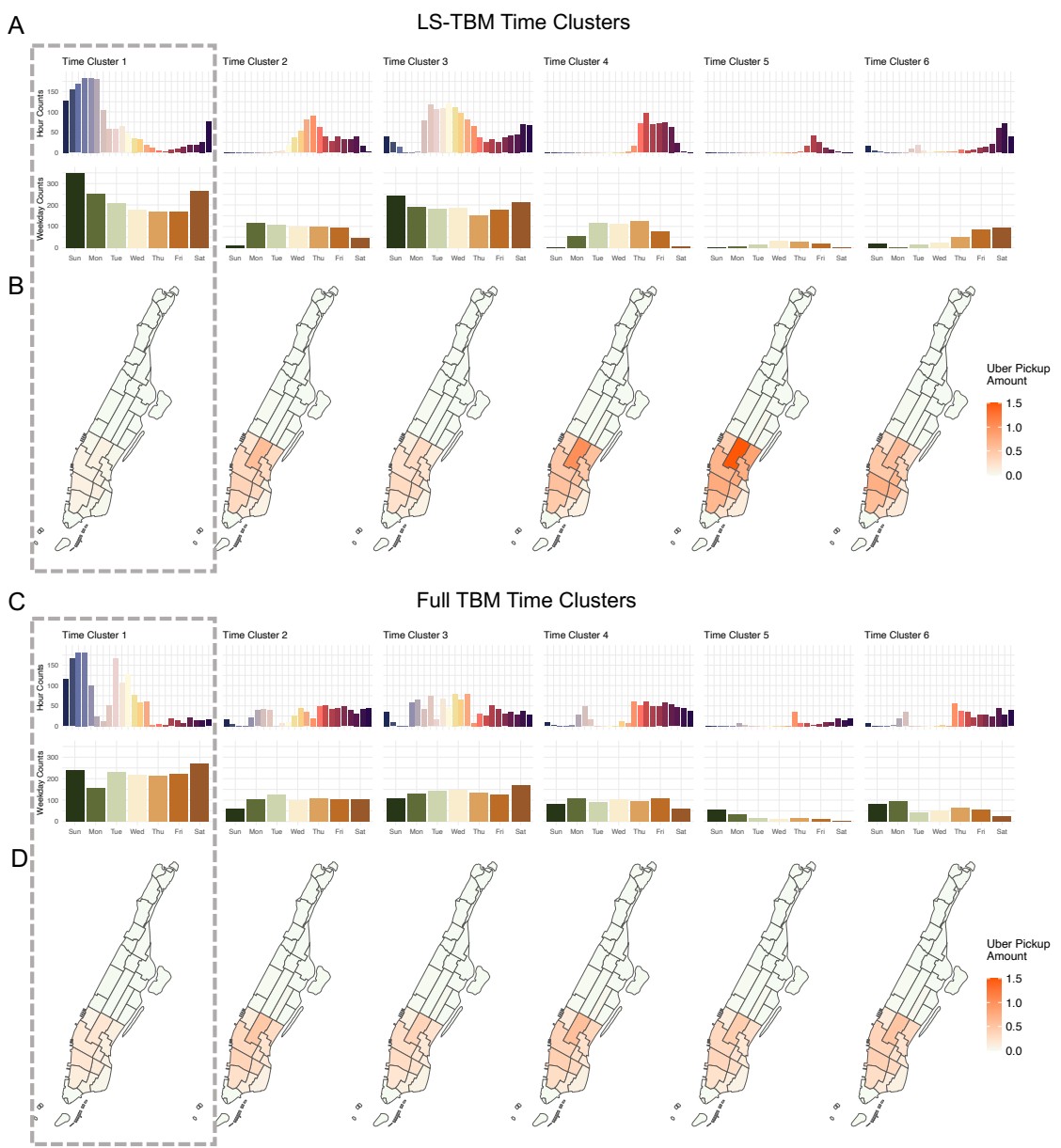

Figure 8: Time cluster results of LS-TBM with $m_1 = 0.1n_1, (m_2, m_3) = 0.5(n_2, n_3)$ and TBM for Uber Pickup data. TBM Time Clusters are relabeled for better comparison. Subplots **A** and **C** report the counts of 24 hours and 7 weekdays for the timestamps in each time cluster. In the hour barplots, the x-axis ranges from 0 to 24. Subplots **B** and **D** visualize the estimated hourly Uber pickup amount (estimated $\mathcal{C}$ of (1)) in Manhattan neighborhoods associated with 6 time clusters. For both LS-TBM and TBM results, Time Cluster 1 (gray dashed rectangles) has the largest group size and the smallest averaged pickup amount.

# C. Proofs

In proofs, for convenience, we use a mixture of notations $\boldsymbol{A}(j,:)$ and $\boldsymbol{A}_{j:}$ to denote the $j$-th row of matrix $\boldsymbol{A}$. Similar notations are used for the rows of a matrix.

## C.1. Proof of Theorem 3.1

*Proof of Theorem 3.1.* Without loss of generality, we assume that $\sigma = 1$, the permutations $\tilde{\pi}_k = \arg\min_{\pi \in \Pi_K} \sum_{i \in S_k} \mathbb{1}\{\tilde{z}_k(i) \neq \pi \circ z_k(i)\}$ are identity mapping on $[r_k]$. We focus on the derivation for the misclassification error on the first mode $k = 1$. We drop the subscript 1 for some matricizations, e.g., $\boldsymbol{Y}, \boldsymbol{C}$, without misunderstanding. The derivations for other modes can be easily extended.

The key proof idea is to decompose the misclassification loss for seeded clustering, $L(\hat{z}_1^c)$, by two parts: the unavoidable statistical loss from the noise, and the inherited loss from the imperfect seeds. Under the model (1), we are able to upper bound the statistical error by the concentration properties of sub-Gaussian variables. For the inherited loss, intuitively, more accurate seeds will lead to less mistakes in the classifications for the rest of nodes. Our main goal is to quantify both statistical and inherited losses to upper bound $L(\hat{z}_1^c)$, and thereof finally obtain the error bound for $\ell(z_1^c)$ with Lemma 1.

We first introduce extra notations for the proof.

- Complement set for seeds:
$$S_k^c = [n_k]/S_k, \quad k \in [K].$$

- Normalized membership matrices for subsets $S_k$'s:
$$\boldsymbol{W}_k \coloneqq \boldsymbol{M}_k(S_k,:)(\operatorname{diag}(\mathbf{1}_{m_k}^T \boldsymbol{M}_k(S_k,:)))^{-1}, \quad \tilde{\boldsymbol{W}}_k \coloneqq \tilde{\boldsymbol{M}}_k(S_k,:)(\operatorname{diag}(\mathbf{1}_{m_k}^T \tilde{\boldsymbol{M}}_k(S_k,:)))^{-1},$$

  where $\tilde{\boldsymbol{M}}_k(S_k,:)$ are membership matrices corresponding to the seed $\tilde{z}_k$, for $k \in [K]$.

- Dual normalized membership matrices:
$$\boldsymbol{V} \coloneqq \boldsymbol{W}_2 \otimes \cdots \otimes \boldsymbol{W}_K, \quad \tilde{\boldsymbol{V}} \coloneqq \tilde{\boldsymbol{W}}_2 \otimes \cdots \otimes \tilde{\boldsymbol{W}}_K,$$

  where $\otimes$ refers to the matrix Kronecker product.

- Sub-tensors corresponding to the subsets $S_k$'s and their matricizations:
$$\begin{aligned}
\mathcal{Y}_S &\coloneqq \mathcal{Y}(S_1,\ldots,S_K), & \mathcal{Y}_{j,S_{-1}} &\coloneqq \mathcal{Y}(j,S_2,\ldots,S_K) \\
\boldsymbol{Y}_S &\coloneqq \operatorname{Mat}_1(\mathcal{Y}_S), & \boldsymbol{Y}_{j,S_{-1}} &\coloneqq \operatorname{Mat}_1(\mathcal{Y}_{j,S_{-1}}).
\end{aligned}$$

  Similar notations are also defined for $\mathcal{X}, \mathcal{E}$.

- Estimation of core tensor with seeds and oracle estimator given true assignments:
$$\tilde{\mathcal{C}} = \mathcal{Y}_S \times_1 \tilde{\boldsymbol{W}}_1^T \times_2 \cdots \times_K \tilde{\boldsymbol{W}}_K^T, \quad \bar{\mathcal{C}} = \mathcal{Y}_S \times_1 \boldsymbol{W}_1^T \times_2 \cdots \times_K \boldsymbol{W}_K^T.$$

  We use matricizations $\tilde{\boldsymbol{C}} = \operatorname{Mat}_1(\tilde{\mathcal{C}})$, $\bar{\boldsymbol{C}} = \operatorname{Mat}_1(\bar{\mathcal{C}})$, and $\boldsymbol{C} = \operatorname{Mat}_1(\mathcal{C})$.

Next, we **decompose** the misclassification loss $L(z_1^c)$ into two parts. Consider an arbitrary node $j \in S_1^c$ with true assignment $z_1(j)$. We consider the key event in which node $j$ is misclassified to cluster $b \neq z_1(j)$:
$$\hat{z}_1(j) = b \quad \Leftrightarrow \quad \|\tilde{\boldsymbol{C}}_{b:} - \boldsymbol{A}_{j:}\|^2 \leq \|\tilde{\boldsymbol{C}}_{z_1(j):} - \boldsymbol{A}_{j:}\|^2, \tag{9}$$

where $\boldsymbol{A}_{j:} = \boldsymbol{Y}_{j,S_{-1}} \tilde{\boldsymbol{V}}$ by the Line 8 in Algorithm 1.

After re-arrangement, the event (9) is equivalent to
$$2\left\langle \boldsymbol{E}_{j,S_{-1}} \boldsymbol{V}, \bar{\boldsymbol{C}}_{z_1(j):} - \bar{\boldsymbol{C}}_{b:}\right\rangle \leq -\|\boldsymbol{C}_{z_1(j):} - \boldsymbol{C}_{b:}\|^2 + F_{jb} + G_{jb} + H_{jb},$$

where

$$F_{jb} = 2\left\langle \boldsymbol{E}_{j,S_{-1}}\tilde{\boldsymbol{V}}, (\bar{\boldsymbol{C}}_{z_1(j):} - \tilde{\boldsymbol{C}}_{z_1(j):}) - (\bar{\boldsymbol{C}}_{b:} - \tilde{\boldsymbol{C}}_{b:})\right\rangle + 2\left\langle \boldsymbol{E}_{j,S_{-1}}(\boldsymbol{V} - \tilde{\boldsymbol{V}}), \bar{\boldsymbol{C}}_{z_1(j):} - \bar{\boldsymbol{C}}_{b:}\right\rangle$$

$$G_{jb} = \left(\|\boldsymbol{X}_{j,S_{-1}}\tilde{\boldsymbol{V}} - \tilde{\boldsymbol{C}}_{z_1(j):}\|^2 - \|\boldsymbol{X}_{j,S_{-1}}\tilde{\boldsymbol{V}} - \boldsymbol{W}_{1,:a}^T\boldsymbol{Y}_S\tilde{\boldsymbol{V}}\|^2\right)$$

$$- \left(\|\boldsymbol{X}_{j,S_{-1}}\tilde{\boldsymbol{V}} - \tilde{\boldsymbol{C}}_{b:}\|^2 - \|\boldsymbol{X}_{j,S_{-1}}\tilde{\boldsymbol{V}} - \boldsymbol{W}_{1,:b}^T\boldsymbol{Y}_S\tilde{\boldsymbol{V}}\|^2\right)$$

$$H_{jb} = \|\boldsymbol{X}_{j,S_{-1}}\tilde{\boldsymbol{V}} - \boldsymbol{W}_{1,:a}^T\boldsymbol{Y}_S\tilde{\boldsymbol{V}}\|^2 - \|\boldsymbol{X}_{j,S_{-1}}\tilde{\boldsymbol{V}} - \boldsymbol{W}_{1,:b}^T\boldsymbol{Y}_S\tilde{\boldsymbol{V}}\|^2 + \|\boldsymbol{C}_{z_1(j):} - \boldsymbol{C}_{b:}\|^2.$$

Then, we are able to upper bound the indicator as

$$\mathbb{1}\{\hat{z}_1(j) = b\} \le \mathbb{1}\{\hat{z}_1(j) = b, \left\langle \boldsymbol{E}_{j,S_{-1}}\boldsymbol{V}, \bar{\boldsymbol{C}}_{z_1(j):} - \bar{\boldsymbol{C}}_{b:}\right\rangle \le -\frac{1}{4}\|\boldsymbol{C}_{z_1(j):} - \boldsymbol{C}_{b:}\|^2\}$$

$$+ \mathbb{1}\{\hat{z}_1(j) = b, \frac{1}{2}\|\boldsymbol{C}_{z_1(j):} - \boldsymbol{C}_{b:}\|^2 \le F_{jb} + G_{jb} + H_{jb}\}.$$

Further, we can upper bound the misclassification loss

$$L(\hat{z}_1^c) \le \xi_1 + \frac{1}{n_1 - m_1}\sum_{j\in S_1^c}\sum_{b\in[r_1]}\zeta_{jb},$$

where

$$\xi_1 = \frac{1}{n_1 - m_1}\sum_{j\in S_1^c}\sum_{b\in[r_1]}\mathbb{1}\{\hat{z}_1(j) = b, \left\langle \boldsymbol{E}_{j,S_{-1}}\boldsymbol{V}, \bar{\boldsymbol{C}}_{z_1(j):} - \bar{\boldsymbol{C}}_{b:}\right\rangle \le -\frac{1}{4}\|\boldsymbol{C}_{z_1(j):} - \boldsymbol{C}_{b:}\|^2\}\|\boldsymbol{C}_{z_1(j):} - \boldsymbol{C}_{b:}\|^2,$$

and

$$\zeta_{jb} = \mathbb{1}\{\hat{z}_1(j) = b, \frac{1}{2}\|\boldsymbol{C}_{z_1(j):} - \boldsymbol{C}_{b:}\|^2 \le F_{jb} + G_{jb} + H_{jb}\}\|\boldsymbol{C}_{z_1(j):} - \boldsymbol{C}_{b:}\|^2.$$

Here, $\xi_1$ is the **statistical loss** due to the existence of noise $\mathcal{E}$. In contrast, the term $\zeta_{jb}$ is controlled by the difference between the oracle estimator $\bar{\mathcal{C}}$ and the estimator based on seeds $\tilde{\mathcal{C}}$, which relies on the accuracy of seeds. The sum of $\zeta_{jb}$ is the **inherited loss**, and should be bounded by the losses of seeds $L(\tilde{z}_k)$'s.

Indeed, Lemmas 6 and 5 provide the upper bounds of $\xi_1$ and the sum of $\zeta_{jb}$. We have

$$L(\hat{z}_1^c) \le \exp\left(-\frac{c_2 m_{-1}}{r_{-1}}\Delta_{\min}^2\right) + c_1 L(\hat{z}_1^c) + cC_1\sqrt{\bar{m}}\ell(\hat{z}_1^c)\sum_{k\in[K]}L(\tilde{z}_k), \tag{10}$$

with high probability $1 - \exp(-c_3 m_1) - \exp(-c_4 m_{-1}\Delta_{\min}^2)$ and positive constants $c$ in condition (3), $c_1 \in (0,1)$, and $c_2, c_3, C_1 \in \mathbb{R}_+$.

By Lemma 1, we have $\ell(z_1^c) \le L(z_1^c)/\Delta_{\min}^2$. Dividing $\Delta_{\min}^2$ on both sides of the inequality (10) and rearranging the terms, we have

$$\ell(\hat{z}_1^c) \le \frac{L(\hat{z}_1^c)}{\Delta_{\min}^2} \le C_2\Delta_{\min}^{-2}\exp\left(-\frac{c_2 m_{-1}}{r_{-1}}\Delta_{\min}^2\right) + cC_1\ell(\hat{z}_1^c)\frac{\sqrt{\bar{m}}\sum_{k\in[K]}L(\tilde{z}_k)}{\Delta_{\min}^2}$$

$$\le C_2\Delta_{\min}^{-2}\exp\left(-\frac{c_2 m_{-1}}{r_{-1}}\Delta_{\min}^2\right) + Kc^2C_1\ell(\hat{z}_1^c).$$

When $c$ is small enough such that $Kc^2C_1 < 1$, we finally have

$$\ell(\hat{z}_1^c) \le C'\Delta_{\min}^{-2}\exp\left(-\frac{c' m_{-1}}{r_{-1}}\Delta_{\min}^2\right),$$

with probability at least $1 - \exp(-c''\underline{m}) - \exp(-c''' m_{-1}\Delta_{\min}^2)$.

$$\square$$

## C.2. Useful lemmas for the proof of Theorem 3.1

In this section, we first list some intermediate lemmas (Lemmas 2, 3, 4) for sub-Gaussian concentration properties and secondary conclusions for misclassification loss analysis. Then, we state and prove the main lemmas (Lemmas 6 and 5) for the upper bounds of statistical and inherited losses.

The intermediate lemmas are similar with the lemmas for Han et al. [9, Proof of Theorem 2] but with different dimension and subject considerations. For clearness, we only show the intermediate results directly used in Lemmas 6 and 5 and highlight the necessary adjustments in the proof. Full lemma statements and proofs can be found in Han et al. [9, Proof of Theorem 2].

Recall that we use $C, C_0, C_1, \ldots$ and $c, c_0, c_1, \ldots$ for generic large and small positive constants, respectively.

**Lemma 1** (Misclassification error and loss). *Consider the misclassification error and loss for $\hat{z}_k$ under the TBM* (1). *We have*

$$\ell(\hat{z}_k) \leq L(\hat{z}_k)/\Delta_{\min}^2, \quad k \in [K].$$

**Lemma 2** (Sub-Gaussian concentration). *Under the set up of Theorem 3.1, as $m_k \to \infty, k \in [K]$, for any $j \in S_1^c$, we have*

$$\|\boldsymbol{E}_{j,S_{-1}}\boldsymbol{V}\| \lesssim \sqrt{\frac{r_{-1}}{m_{-1}}}(1 + \sqrt{r_{-1}}). \tag{11}$$

*Proof of Lemma 2.* Lemma 2 is the Corollary of Han et al. [9, Lemma 9]. □

**Lemma 3** (Upper bound for membership matrix difference). *Under the set up of Theorem 3.1 and Lemma 2, as $m_k \to \infty, k \in [K]$, we have*

$$\|\boldsymbol{E}_{j,S_{-1}}(\boldsymbol{V} - \tilde{\boldsymbol{V}})\| \lesssim \sqrt{\frac{r_{-1}(r_{-1} + \bar{m}\bar{r})}{m_{-1}}} \sum_{k=2}^{K} \frac{r_k L(\tilde{z}_k)}{\Delta_k^2}.$$

*Proof of Lemma 3.* See Han et al. [9, Lemma 5]. □

**Lemma 4** (Upper bounds for $F_{jb}, G_{jb}, H_{jb}$). *Under the set up of Theorem 3.1, as $m_k \to \infty, k \in [K]$, we have*

$$\max_{j \in S_1^c} \max_{b \neq z_1(j)} \frac{F_{jb}^2}{\|\boldsymbol{C}_{z_1(j):} - \boldsymbol{C}_{b:}\|^2} \lesssim \frac{\sum_{k=1}^{K} r_k L(\tilde{z}_k)}{\Delta_1^2}\|\boldsymbol{E}_{j,S_{-1}}\boldsymbol{V}\|^2$$

$$+ \left(1 + \frac{\sum_{k=1}^{K} r_k L(\tilde{z}_k)}{\Delta_1^2}\right)\|\boldsymbol{E}_{j,S_{-1}}(\boldsymbol{V} - \tilde{\boldsymbol{V}})\|^2, \tag{12}$$

$$\max_{j \in S_1^c} \max_{b \neq z_1(j)} \frac{G_{jb}^2}{\|\boldsymbol{C}_{z_1(j):} - \boldsymbol{C}_{b:}\|^2} \leq c_0\left(\Delta_1^2 + \sum_{k=1}^{K} L(\tilde{z}_k)\right), \tag{13}$$

$$\max_{j \in S_1^c} \max_{b \neq z_1(j)} \frac{|H_{jb}|}{\|\boldsymbol{C}_{z_1(j):} - \boldsymbol{C}_{b:}\|^2} \leq \frac{1}{4}, \tag{14}$$

*where $c_0 < \frac{1}{128K}$ is some small constant.*

*Proof of Lemma 4.* Notice that by Assumption 2, there exists some $i \in S_1$ such that $z_1(i) = z_1(j)$ and thus $\boldsymbol{X}_{j,S_{-1}} = \boldsymbol{X}_{i,S_{-1}}$. Therefore, we are able to upper bound $F_{jb}, G_{jb}, H_{jb}$ following the proofs of inequalities (75), (76), and (77) in [9]. □

**Lemma 5** (Upper bound for the sum of $\zeta_{jb}$). *Under the set up of Theorem 3.1, we have*

$$\frac{1}{n_1 - m_1} \sum_{j \in S_1^c} \sum_{b \in [r_1]/z_1(j)} \zeta_{jb} \leq c_1 L(\hat{z}_1^c) + cC_1\sqrt{\bar{m}}\ell(\hat{z}_1^c) \sum_{k \in [K]} L(\tilde{z}_k)$$

*for some constants $c_1 \in (0, 1)$ and $C_1 > 0$, and $c$ is the small constant in condition* (3).

*Proof of Lemma 5.* Note that

$$
\zeta_{jb} = \mathbb{1}\{\hat{z}_1(j) = b, \frac{1}{2}\|C_{z_1(j):} - C_{b:}\|^2 \le F_{jb} + G_{jb} + H_{jb}\}\|C_{z_1(j):} - C_{b:}\|^2
$$

$$
\le \mathbb{1}\{\hat{z}_1(j) = b, \frac{1}{4}\|C_{z_1(j):} - C_{b:}\|^2 \le F_{jb} + G_{jb}\}\|C_{z_1(j):} - C_{b:}\|^2
$$

$$
\le 64\mathbb{1}\{\hat{z}_1(j) = b\}\left(\frac{F_{jb}^2}{\|C_{z_1(j):} - C_{b:}\|^2} + \frac{G_{jb}^2}{\|C_{z_1(j):} - C_{b:}\|^2}\right),
$$

where the first inequality follows from inequality (14) in Lemma 4, and the last inequality follows by the fact that $\mathbb{1}\{1 \le x\} \le x^2$ for $x \in \mathbb{R}$.

We first consider the summation over $F_{jb}^2$. With inequality (12) in Lemma 4, we have

$$
\frac{1}{n_1 - m_1}\sum_{j \in S_1^c}\sum_{b \in [r_1]/z_1(j)}\mathbb{1}\{\hat{z}_1(j) = b\}\frac{F_{jb}^2}{\|C_{z_1(j):} - C_{b:}\|^2}
$$

$$
\le \frac{1}{n_1 - m_1}\sum_{j \in S_1^c}\mathbb{1}\{\hat{z}_1(j) \ne z_1(j)\}\max_{b \ne z_1(j)}\frac{F_{jb}^2}{\|C_{z_1(j):} - C_{b:}\|^2}
$$

$$
\le F_1 + F_2, \tag{15}
$$

where

$$
F_1 = \frac{1}{n_1 - m_1}\sum_{j \in S_1^c}\mathbb{1}\{\hat{z}_1(j) \ne z_1(j)\}\frac{\sum_{k=1}^K r_k L(\tilde{z}_k)}{\Delta_1^2}\|E_{j,S_{-1}}V\|^2,
$$

$$
F_2 = \frac{1}{n_1 - m_1}\sum_{j \in S_1^c}\mathbb{1}\{\hat{z}_1(j) \ne z_1(j)\}\left(1 + \frac{\sum_{k=1}^K r_k L(\tilde{z}_k)}{\Delta_1^2}\right)\|E_{j,S_{-1}}(V - \tilde{V})\|^2.
$$

By inequality (11) in Lemma 2, we have

$$
F_1 \lesssim \frac{1}{n_1 - m_1}\sum_{j \in S_1^c}\mathbb{1}\{\hat{z}_1(j) \ne z_1(j)\}\frac{\sum_{k=1}^K r_k L(\tilde{z}_k)}{\Delta_{\min}^2}\frac{r_{-1}^2}{m_{-1}} \lesssim \ell(\hat{z}_1^c)\sum_{k \in [K]}L(\tilde{z}_k), \tag{16}
$$

given condition (3) such that $\Delta_{\min}^2 \gtrsim m_{-1}^{-1}$.

By Lemma 3, we have

$$
F_2 \lesssim \frac{1}{n_1 - m_1}\sum_{j \in S_1^c}\mathbb{1}\{\hat{z}_1(j) \ne z_1(j)\}\left(1 + \frac{\sum_{k=1}^K r_k L(\tilde{z}_k)}{\Delta_1^2}\right)\frac{r_{-1}^2 + \bar{m}\bar{r}}{m_{-1}}\sum_{k=2}^K\frac{r_k^2 L^2(\tilde{z}_k)}{\Delta_k^4}
$$

$$
\lesssim \frac{1}{n_1 - m_1}\sum_{j \in S_1^c}\mathbb{1}\{\hat{z}_1(j) \ne z_1(j)\}\frac{\sqrt{\bar{m}}}{m_{-1}}\frac{\sum_{k=1}^K L(\tilde{z}_k)}{\Delta_{\min}^2}
$$

$$
\lesssim \sqrt{\bar{m}}\ell(\hat{z}_1^c)\frac{\sum_{k=1}^K L(\tilde{z}_k)}{m_{-1}\Delta_{\min}^2}
$$

$$
\le cC_1\sqrt{\bar{m}}\ell(\hat{z}_1^c)\sum_{k=1}^K L(\tilde{z}_k), \tag{17}
$$

where the second inequality follows by condition (3) such that $L(\tilde{z}_k)/\Delta_{\min}^2 \le c/\sqrt{\bar{m}}$ for all $k \in [K]$ and the last inequality follows by $\Delta_{\min}^2 \gtrsim m_{-1}^{-1}$.

Next, we consider the summation over $G_{jb}^2$. With inequality (13) in Lemma 4, we have

$$\frac{1}{n_1 - m_1} \sum_{j \in S_1^c} \sum_{b \in [r_1]/z_1(j)} \mathbb{1}\{\hat{z}_1(j) = b\} \frac{G_{jb}^2}{\|\boldsymbol{C}_{z_1(j):} - \boldsymbol{C}_{b:}\|^2}$$

$$\leq \frac{1}{n_1 - m_1} \sum_{j \in S_1^c} \mathbb{1}\{\hat{z}_1(j) \neq z_1(j)\} \max_{b \neq z_1(j)} \frac{G_{jb}^2}{\|\boldsymbol{C}_{z_1(j):} - \boldsymbol{C}_{b:}\|^2}$$

$$\leq \frac{c_0}{n_1 - m_1} \sum_{j \in S_1^c} \mathbb{1}\{\hat{z}_1(j) \neq z_1(j)\} \left( \Delta_1^2 + \sum_{k=1}^K L(\tilde{z}_k) \right)$$

$$\leq c_1 L(\hat{z}_1^c) + c_0 \ell(\hat{z}_1^c) \sum_{k=1}^K L(\tilde{z}_k), \tag{18}$$

for some $c_1 \in (0, 1)$.

Taking $C_1$ large enough, we have term $F_2$ dominants $F_1$ and the second term in inequality (18). Then, combining the upper bounds (15), (16), (17), and (18), we have shown the upper bound for the sum of $\zeta_{jb}$ in Lemma 5. □

**Lemma 6** (Upper bound for $\xi_1$). *Under the set up of Theorem 3.1, we have*

$$\xi_1 \leq \exp\left( -\frac{c_2 m_{-1}}{r_{-1}} \Delta_{\min}^2 \right)$$

*with probability at least* $1 - \exp(-c_3 m_1) - \exp(-c_4 m_{-1} \Delta_{\min}^2)$ *for some small positive constants* $c_2, c_3, c_4$.

*Proof of Lemma 6.* Recall that

$$\mathbb{E}[\xi_1] = \frac{1}{n_1 - m_1} \sum_{j \in S_1^c} \sum_{b \in [r_1]/z_1(j)} \|\boldsymbol{C}_{z_1(j):} - \boldsymbol{C}_{b:}\|^2 \mathbb{P}(\langle e_j, \bar{\boldsymbol{C}}_{z_1(j):} - \bar{\boldsymbol{C}}_{b:} \rangle \leq -\frac{1}{4} \|\boldsymbol{C}_{z_1(j):} - \boldsymbol{C}_{b:}\|^2)$$

where $e_j := \boldsymbol{E}_{j,S_{-1}} \boldsymbol{V}$. Note that $e_j$'s are independent random vectors in $\mathbb{R}^{r_{-1}}$, whose entries are independently sub-Gaussian distributed with norm bounded by $\mathcal{O}(\sqrt{r_{-1}/m_{-1}})$ based on Lemma 2.

We have the upper bound for the probability

$$\mathbb{P}(\langle e_j, \bar{\boldsymbol{C}}_{z_1(j):} - \bar{\boldsymbol{C}}_{b:} \rangle \leq -\frac{1}{4} \|\boldsymbol{C}_{z_1(j):} - \boldsymbol{C}_{b:}\|^2)$$

$$\leq \mathbb{P}(\langle e_j, \boldsymbol{C}_{z_1(j):} - \bar{\boldsymbol{C}}_{b:} \rangle \leq -\frac{1}{8} \|\boldsymbol{C}_{z_1(j):} - \boldsymbol{C}_{b:}\|^2) + \mathbb{P}(\langle e_j, \bar{\boldsymbol{C}}_{z_1(j):} - \boldsymbol{C}_{z_1(j):} \rangle \leq -\frac{1}{16} \|\boldsymbol{C}_{z_1(j):} - \boldsymbol{C}_{b:}\|^2)$$

$$+ \mathbb{P}(\langle e_j, \boldsymbol{C}_{b:} - \bar{\boldsymbol{C}}_{b:} \rangle \leq -\frac{1}{16} \|\boldsymbol{C}_{z_1(j):} - \boldsymbol{C}_{b:}\|^2)$$

$$\leq 5 \exp\left( -\frac{cm_{-1}}{r_{-1}} \|\boldsymbol{C}_{z_1(j):} - \boldsymbol{C}_{b:}\|^2 \right), \tag{19}$$

with probability $1 - \exp(-c_3 m_1)$, where the last inequality follows from Han et al. [9, Lemma 6], replacing $\boldsymbol{S}$ terms by $\boldsymbol{C}$ terms in our context. Note that unlike Han et al. [9, Lemma 6], $e_j$ is independent with $\boldsymbol{C}_{b:} - \bar{\boldsymbol{C}}_{b:}$ for all $j \in S_1^c, b \in [r_1]$ since the randomness are from two non-overlapped parts of the noise tensor. The inner produce $\langle e_j, \bar{\boldsymbol{C}}_{z_1(j):} - \boldsymbol{C}_{z_1(j):} \rangle$ turns out to be the weighted sum of products of two independent random vectors $e_j^T e_l$ for $j \in S_1^c, l \in S_1$. Since Han et al. [9, Lemma 6] merely relies on the Bernstein inequality for $e_j^T e_l$ with $j \neq l$, Lemma 6 also works for our case and inequality (19) holds.

Hence, we have upper bound of the expectation

$$\mathbb{E}[\xi_1] \leq \frac{5}{n_1 - m_1} \sum_{j \in S_1^c} \sum_{b \in [r_1]/z_1(j)} \|C_{z_1(j):} - C_{b:}\|^2 \exp\left(-\frac{cm_{-1}}{r_{-1}} \|C_{z_1(j):} - C_{b:}\|^2\right)$$

$$\leq \exp\left(-\frac{cm_{-1}}{2r_{-1}} \|C_{z_1(j):} - C_{b:}\|^2\right)$$

$$\leq \exp\left(-\frac{cm_{-1}}{2r_{-1}} \Delta_{\min}^2\right).$$

With Markov inequality, we have

$$\mathbb{P}\left(\xi_1 \leq \mathbb{E}[\xi_1] + \exp\left(-\frac{cm_{-1}}{4r_{-1}} \Delta_{\min}^2\right)\right) \geq 1 - \exp\left(-\frac{c_3 m_{-1}}{4r_{-1}} \Delta_{\min}^2\right).$$

Then, we have finished the proof of Lemma 6.

$\square$

## C.3. Proof of Theorem 3.2

*Proof of Theorem 3.2.* Without loss of generality, we only prove the sub-tensor balance property on the first mode $k = 1$. Then, we drop the subscript 1 from $n_1, m_1, r_1, S_1, z_1$. The balance property on other modes can be proved by modifying the subscripts.

We first define the number of nodes in $a$-th community and the ratio of nodes in $a$-th community:

$$n_a = \sum_{i \in [n]} \mathbb{1}\{z(i) = a\}, \quad m_a = \sum_{i \in S} \mathbb{1}\{z(i) = a\}, \quad \mu_a = n_a/n.$$

By random sampling, $m_a$ follows Hypergeometric distribution with parameters $(m, n_a, n)$ for all $a \in [r]$. Notice that $m_a$'s are not independent but the conclusion for marginal distributions holds.

By Corollary 1 in [40], for arbitrary $a \in [r]$, we have

$$\mathbb{P}(m_a \geq m\mu_a + mt) \leq \exp\left(-\frac{mt^2}{\sigma_a^2(1 - \frac{m-1}{m-1}) + \frac{t}{3}}\right) \leq \exp\left(-\frac{mt^2/2}{\mu_a + \frac{t}{3}}\right), \quad \text{for } t > 0, \qquad (20)$$

where $\sigma_a^2 = \mu_a(1 - \mu_a)$ and the second inequality holds by $\mu_a < 1$.

Similarly, by symmetry, we have

$$\mathbb{P}(m_a \leq m\mu_a - mt) \leq \exp\left(-\frac{mt^2}{\sigma_a^2(1 - \frac{m-1}{m-1}) + \frac{t}{3}}\right) \leq \exp\left(-\frac{mt^2/2}{\mu_a + \frac{t}{3}}\right), \quad \text{for } t > 0.$$

Take $t = \frac{1}{4}\mu_a$. We have upper bound for the maxima of $m_a$

$$\mathbb{P}\left(\max_{a \in [r]} m_a \leq \frac{5}{4} m \max_{a \in [r]} \mu_a\right) = \mathbb{P}\left(\bigcap_{a \in [r]} \{m_a \leq \frac{5}{4} m \max_{a \in [r]} \mu_a\}\right)$$

$$\geq \mathbb{P}\left(\bigcap_{a \in [r]} \{m_a \leq \frac{5}{4} m\mu_a\}\right)$$

$$\geq 1 - \sum_{a \in [r]} \exp\left(-\frac{3m\mu_a}{128}\right)$$

$$\geq 1 - r \exp\left(-\frac{3m\alpha_1}{128r}\right), \qquad (21)$$

where the second inequality follows by the union bound and tail bound (20), and the last inequality holds with Assumption 1 indicating $\frac{\alpha_1}{r} \leq \mu_a \leq \frac{\alpha_2}{r}$.

Similarly, we have lower bound for the minima of $m_a$

$$\mathbb{P}\left(\min_{a \in [r]} m_a \geq \frac{3}{4} m \min_{a \in [r]} \mu_a\right) \geq 1 - r \exp\left(-\frac{3m\alpha_1}{128r}\right). \tag{22}$$

Finally, take $\alpha_3 < \frac{3\alpha_1}{4}$ and $\alpha_4 > \frac{5\alpha_2}{4}$. We finish the proof of Theorem 3.2 by combining inequalities (21) and (22):

$$\mathbb{P}\left(\frac{\alpha_3 m}{r} \leq m_a \leq \frac{\alpha_4 m}{r}, \text{ for all } a \in [r]\right) = \mathbb{P}\left(\frac{\alpha_3 m}{r} \leq \min_{a \in [r]} m_a \text{ and } \max_{a \in [r]} m_a \leq \frac{\alpha_4 m}{r}\right)$$

$$\geq \mathbb{P}\left(\min_{a \in [r]} m_a \geq \frac{3}{4} m \min_{a \in [r]} \mu_a \text{ and } \max_{a \in [r]} m_a \leq \frac{5}{4} m \max_{a \in [r]} \mu_a\right)$$

$$\geq 1 - 2r \exp\left(-\frac{3m\alpha_1}{128r}\right).$$

$\square$

