# OpenReview forum: "Large-Scale Multiway Clustering with Seeded Clustering"
_CPAL.cc/2025/Proceedings_Track — CPAL 2025 (Proceedings Track) Poster_

### Official Review · Reviewer_72gW · 2025-01-07
**Good speed-ups while preserving accuracy**

**Rating:** 8
**Confidence:** 3

**Review:**

This paper introduces a scalable method to cluster tensor data with theoretical analyses and good empirical results

Pros:
- clear diagram of algorithm
- significant improvement on speed with minimal change in error

Cons:
- validation on more than just one real dataset wold be good to include
- lack of empirical exploration on higher order tensors

Questions:
1) Empirically, what is the relationship between the runtime and the order of the tensor (similarly with error vs order)?
2) What happens to the recovery if instead of randomly sampling a sub-tensor, we sampled in a more structured manner (e.g. grid sampling, mean pooling, etc)?

---

### Official Review · Reviewer_cSjM · 2025-01-08
**Large-Scale Multiway Clustering with Seeded Clustering**

**Rating:** 6
**Confidence:** 1

**Review:**

I am not engaged in research related to this paper, so I am unable to provide an objective
evaluation on this topic. Please disregard my review comments.

---

### Official Review · Reviewer_r1PJ · 2025-01-10
**Review of submission 78**

**Rating:** 7
**Confidence:** 3

**Review:**

**Strengths**

1. The topic of the paper, improving efficiency of multi-way clustering methods to enable more widespread adoption, nicely fits the goals of  CPAL.

2. The developed method is simple, but has provable guarantees and leads to significant reduction of costs in synthetic and real-world data sets.

3. I really liked the application of LS-TBM to the Uber pickup data set. The extracted modes were interpretable (and provided parsimony) and seeing the developed method in action on such a big data set (of industry application) was nice.

**Weaknesses**

1. While the LS-TBM learns modes that "make sense" (Fig. 5B) and the overall clustering is similar between the full TBM and LS-TBM (Fig. 5B), TBM and LS-TBM learn different modes (Fig. 6 vs. Fig 7). This should be discussed in the main text. Given that tensor methods are usually stochastic, I imagine these differences could (in part) be explained by the fact that each time you run TBM and LS-TBM you get different modes. However, it would be nice to demonstrate this. Could you run the full TBM multiple times (I know it is expensive, but even 3 times) and see how much variability there is on each run in the modes, and then compare that to the difference between the TBM and LS-TBM modes. It would be nice to demonstrate on this real-world data set that use of LS-TBM is not getting you something different from TBM. Also, discussion on how tensor methods can get you different mode decompositions on each run might be helpful in giving readers a more balanced perspective on these methods.

2. I liked that the authors numerically tested their theoretical results in Fig. 3. However, it was not entirely clear to me that, in the case of weak SNR (Fig. 3 right), the use of $m=O(n^{1/2})$ was really optimal. Indeed, the CER diverges pretty significantly at $n > 400$ and $n / 4$ seems to do a lot better. Discussing why this is and what it implies for the theoretical results would be good.

3. The figure labels could contain a little more information. For instance, it was not clear to me how Fig. 2 demonstrates a phase transition. Providing more discussion of this would be helpful, especially for figures that are not referenced elsewhere too much.

**Minor points**

1. There are some typos throughout the paper, primarily in the Related work section. Egs. "First line is..." (line 57) should be "The first line is..." and "Though, a few works consider..." (line 71) should be "A few works consider...".

---

### Meta-Review · Area_Chair_UAMt · 2025-02-04

**Recommendation:** Accept (Poster)
**Confidence:** 5

**Metareview:**

This paper presents an efficient way to perform large-scale multiway clustering using a tensor block model (TBM).  The paper is well structured and the proposed approach comes with theoretical guarantees that shed light on important of the proposed approach. The authors demonstrate the merits of the approach in terms of computational & memory efficiency and performance accuracy of the method on both synthetic and real data. During the rebuttal phase, the authors addressed all reviewers' comments. Overall, the paper makes nice contributions on a relevant topic of research and thus I recommend its acceptance to CPAL.

---

### Decision · Program_Chairs · 2025-02-11

Accept (Poster)